# Single-molecule long-read sequencing reveals a conserved intact long RNA profile in sperm

Yu H. Sun [1,2,7], Anqi Wang[3,7], Chi Song [4,6], Goutham Shankar [1], Rajesh K. Srivastava[5], Kin Fai Au [3✉] & Xin Zhiguo Li [1,2✉]

Sperm contributes diverse RNAs to the zygote. While sperm small RNAs have been shown to impact offspring phenotypes, our knowledge of the sperm transcriptome, especially the composition of long RNAs, has been limited by the lack of sensitive, high-throughput experimental techniques that can distinguish intact RNAs from fragmented RNAs, known to abound in sperm. Here, we integrate single-molecule long-read sequencing with short-read sequencing to detect sperm intact RNAs (spiRNAs). We identify 3440 spiRNA species in mice and 4100 in humans. The spiRNA profile consists of both mRNAs and long non-coding RNAs, is evolutionarily conserved between mice and humans, and displays an enrichment in mRNAs encoding for ribosome. In sum, we characterize the landscape of intact long RNAs in sperm, paving the way for future studies on their biogenesis and functions. Our experimental and bioinformatics approaches can be applied to other tissues and organisms to detect intact transcripts.

[1] Center for RNA Biology: From Genome to Therapeutics, University of Rochester Medical Center, Rochester, NY, USA. [2] Department of Biology, University of Rochester, Rochester, NY, USA. [3] Department of Internal Medicine, University of Iowa, Iowa City, IA, USA. [4] College of Public Health, Division of Biostatistics, The Ohio State University, Columbus, OH, USA. [5] Department of Obstetrics/Gynecology, University of Rochester Medical Center, Rochester, NY, USA. [6] Present address: Division of Reproductive Endocrinology, Geisinger Medical Center, Danville, PA, USA. [7] These authors contributed equally: Yu H. Sun, Anqi Wang. ✉email: kinfai.au@osumc.edu; Xin_Li@urmc.rochester.edu

Sperm contribute DNA and diverse species of RNA to the zygote during fertilization[1–4]. While the sperm RNA contribution is relatively small compared to that of oocyte RNAs, sperm RNAs have been shown to mediate epigenetic inheritance[5–11]. This form of inheritance has been well established in plants, fungi, worms, and flies[12], yet, we are only beginning to understand sperm RNA-mediated inheritance in mammals. A recent analysis of obese patients and lean controls identified differences in sperm small RNAs, which could be partially reset in response to aggressive weight loss[5]. Another study identified the differential expression of long non-coding RNAs (lncRNAs) between diabetic and non-diabetic mice[6]. Environmental effects caused by early mental stress, chronic stress[7–9], or a low protein/high fat diet during adolescence[10,11] have been shown to be passed to the next generation through miRNAs or tRNA fragments in sperm. Thus, sperm RNA-mediated epigenetic inheritance likely contributes to inherited disorders associated with gene-environment interactions, including many metabolic diseases[13,14] and neurodevelopmental and neuropsychiatric disorders[15,16].

While sperm small RNAs have been well-characterized[7,10,11], we know little about sperm long RNAs (>200nt), including mRNAs and lncRNAs. When the meiotic products, round spermatids, package their genomes into compacted nuclei in preparation for delivery to oocytes, a stage known as spermiogenesis, the spermatid genome enters a transcriptionally quiescent state[17]. The majority of cytoplasm in round spermatids is enclosed within cytoplasmic vesicles called residual bodies, which are engulfed by Sertoli cells and degraded[18]. 80 S ribosomes are removed as a component of the residual body[19]. As cytosolic translation has ceased in sperm, intact long RNAs are thought to be not only unnecessary for sperm maturation, but also a burden for the cargo-carrying function of sperm. It is thus posited that most sperm long RNAs are unlikely to be functional.

As a consequence of losing 80 S ribosomes, only fragmented rRNAs can be detected in sperm[20]. Total RNA in sperm has a size distribution lacking the 18 S and 28 S rRNA peaks[21] (Supplementary Fig. 1a), reminiscent of degraded RNA samples, the observation of which has led to the wide acceptance of the hypothesis that most long RNAs undergo elimination during spermiogenesis via an unidentified RNase, leaving degraded products in sperm[20,22]. Given most sperm long RNAs are thought to be degraded, research has been biased towards small RNAs, with sperm RNA analysis being focused on exon fragments rather than transcriptional units[23]. Recently, the discovery of sperm long RNA-mediated epigenetic inheritance, and the loss of the transgenerational effect with fragmented long RNAs[24], suggest that sperm long RNAs can not only be modulated by environmental conditions, but also impact the phenotypes of offspring, in their intact form. We therefore decided to revisit the long-standing question of whether intact long RNAs exist in sperm.

The study of sperm long RNAs has been limited by the lack of sensitive, high-throughput techniques that can distinguish intact RNAs from fragmented RNAs in sperm, and that can define the complete sperm transcriptome. Northern blotting can reveal the intactness of mRNAs, but requires significant quantities of input RNA (micrograms), and is low throughput. High-throughput RNA sequencing by Illumina technology (short-read/second-generation sequencing) can only attain short contiguous read lengths (at most 300 bp). Thus, library preparations result in fragmented RNAs, causing transcript integrity loss. Studies based on short-read sequencing have been performed on sperm RNAs[20,23,25,26], but it has not been established whether and to what extent intact mRNAs exist in sperm.

To test the intactness of sperm long RNAs, we also need precise annotation of the transcriptome in testis and sperm. However, the reference transcriptome (RefSeq) annotation is incomplete and unprecise. Testes, where germ cells (precursors of sperm) constitute ~90% of the cell population[27,28], usually express specific RNA isoforms that cannot be found in other tissues[29–31]. The short-read length, loss of linkage information in intron-exon structures, and uncertain source of the multi-mapping reads lead to a failure in accurately defining transcript structures (e.g., 5′ and 3′ boundaries and splicing diversity), and to a failure in discovering unannotated transcripts harboring repetitive sequences. Although attempts have been made to assemble transcriptomes based on short reads[32,33], the lack of information on transcript integrity suggests the problem is unlikely to be overcome computationally.

Single-molecule long-read sequencing technology (third-generation sequencing; PacBio Iso-Seq, Supplementary Fig. 1b) provides a means to identify full-length transcripts[34]. The PacBio system can produce long reads (up to 30 kb vs 250 bp for short-read sequencing). Although long-read sequencing overcomes many of the problems of short-read sequencing, there are still several limitations. The corresponding high error rate in base calling (10–15%) can be reduced by self-error correction, by increasing sequencing depth or circular consensus sequencing/CCS, or by hybrid error correction, using high-quality short reads. Even with correction, determining the accurate 5′- and 3′-boundaries of long reads remains challenging[35,36]. Moreover, Iso-Seq cannot distinguish intact transcripts with an N(5′)ppp(5′)N cap structure (5′-cap) from decay intermediates with a 5′-monophosphate or a 5′-hydroxyl. As revealed in this study, the use of oligo-dT primers to capture mRNA polyA tails in Iso-Seq (Supplementary Fig. 1b) can produce off-target effects by priming from adenine runs within transcripts (Supplementary Fig. 1c). Thus, there is a need for complementary data and computational methods to address these intrinsic problems associated with the reconstitution of the transcriptome.

Here, we present a comprehensive characterization of sperm transcriptomes by pairing PacBio Iso-Seq sequencing with Illumina sequencing of 5′-ends bearing a 5′cap (cap analysis of gene expression; CAGE[37]) and the 3′-ends preceding the poly(A) tail (polyadenylation site sequencing; PAS-Seq[38]). In total, we identified 3440 intact transcript species from 1624 genes in mouse sperm, and 4100 intact transcript species from 2205 genes in human sperm. The spiRNA profile is conserved in mice and humans, and functionally enriched for translation machinery. In sum, our study reveals that intact long RNAs exist in sperm. The identification of the conserved spiRNA profile expands the potential information carried by sperm and provides a short-list of RNAs that can be used for diagnostic purposes. Our integrated computational pipeline for identifying intact transcripts and characterizing accurate transcriptomes in sperm and testis provides a valuable resource for studies beyond male reproduction.

## Results

**Acquisition of ultra-pure sperm for RNA preparation in mice.** Given that a sperm contains RNA in the femto-gram range[39–45], and a typical mammalian cell contains 10–30 pg RNA, sperm purification is critical to avoid somatic RNA contamintation[23,46,47]. To collect sperm with high purity, we combined a swim-up procedure with a somatic cell lysis procedure, each of which has been used independently. We collected sperm by letting them swim out of cauda epididymis, and then performed a hypotonic treatment followed by a mild detergent treatment to remove contaminating somatic cells[21,48]. The somatic lysis treatment buffer and treatment time were optimized to ensure that sperm integrity was not affected[49]. We quantified the somatic cell contamination after the swim-up procedure and

detected 0.3% somatic cells that disappeared after the somatic lysis procedure (Supplementary Fig. 1d, Supplementary Table 1). We further analyzed thousands of sperm under the microscope: we not only confirmed their purity, but also found that the cytoplasmic droplets (known as Hermes bodies), which are shed during post-testicular maturation[50,51], were also efficiently removed in the purification process (Supplementary Fig. 1e). Considering that the lesion in the cauda made from the swim-up procedure could release somatic RNAs that may attach to sperm[52,53], we measured the abundance of *Myh11*, a marker for epididymis contamination[54] that is both broadly and abundantly expressed in the muscular sheath of the cauda epididymis[55], to assess the purity of the sperm RNA samples. We purified the total RNA from the cauda that we dissected for the swim-up procedure and performed a serial dilution for quantitative qPCR analysis. The presence of *Myh11* in our sperm RNA samples was below the detection limit (<1/10,000, Supplementary Table 2). We further took advantage of the recently published single-cell RNA-seq data of epididymis, which contains 21 types of epididymal somatic cells[55], and performed a linear regression to test whether these epididymal somatic cells could explain the sperm expression profiles. We found that no linear combination of the expression profiles of 21 epididymal somatic cells could describe any significant proportion of the expression profiles of our sperm sample ($R^2$ of linear regression = 0.019, Supplementary Data 1). That is, there is very little or no mixture (i.e., contamination) of epididymal somatic cells in our sperm sample. Based on these data, we considered our sperm RNA to be ultra-pure.

**Characterization of the mouse sperm transcriptome**. We generated 256,897 PacBio Iso-Seq long reads to identify the full-length transcripts in purified mouse sperm (Supplementary Data 2), with the depth reaching saturation for isoform identification (Supplementary Fig. 1f). To avoid sequencing-method length biases, we separated the cDNA libraries based on length, and sequenced them separately. We also sequenced the adult mouse testis and analyzed it together with the sperm sample for two reasons. First, it serves a positive control for PacBio Iso-Seq sequencing. Second, the low yield of sperm RNAs limits the feasibility to generate CAGE and PAS libraries from sperm. Given that most sperm RNAs are carried over from spermatogenesis, sperm RNAs can be considered as a subset of testicular RNAs. Thus, CAGE and PAS libraries from testis can be used to correct the pool of intact transcripts from sperm.

Illumina short reads from mouse sperm were used to assemble transcripts and correct the single base errors within the PacBio long reads (Fig. 1, step i). The corrected long reads were aligned to the short-read assembled transcripts, and those supported by corrected long reads were used for further analysis (Fig. 1, step ii). Long reads not supported by assembled transcripts were 'rescued' (Fig. 1, step iv) as they may represent intact transcripts that the short-read based methods have failed to assemble due to sequence bias or repetitive regions (see Fig. 2a). We used our published high-throughput CAGE data from mouse testes[56] to identify intact RNAs and refine the transcript 5′-end boundaries (Fig. 1, step iii). Also, we used our published PAS-Seq data from mouse testis[56] to filter out internally primed artifactual transcripts from the final set of verified intact RNAs (Fig. 1, step v).

**Intact long RNAs are present in mouse sperm**. Our study reveals the existence of intact long RNA species in mouse sperm. Based on the PacBio Iso-Seq data from sperm, we detected 3,440 spiRNA species from 1,624 gene loci. Consistent with the identification of novel isoforms when applying long-read sequencing in previous studies[34,35,57–59], only 755 transcript

species were exactly the same as those deposited in RefSeq whereas 198 transcript species were from loci not included in RefSeq, 7 transcripts were anti-sense transcripts to known loci (see Fig. 2a, Supplementary Fig. 2a, for examples) and 2,479 transcript species were novel isoforms from annotated loci (Fig. 2b, Supplementary Fig. 2b). Tissue specific isoforms mainly arose from the use of alternative PolyA sites (APAs), with smaller numbers arising from alternative splicing or alternative transcription initiation (Fig. 2c). Only one spiRNA was found to span two neighboring genes, *Pate6* and *Pate7*, which are separated by LINE elements (Fig. 2d). Since these genes are mouse specific transcripts lacking orthologs to confirm their unique identity, we treated the previous annotated genes as two isoforms of one gene locus. We have validated the transcripts via Oxford Nanopore Technologies (ONT), an alternative single-molecule long-read sequencing method[35]. Of all isoforms in the mouse assembly, 1,115 had at least one full-length ONT long read. We have also validated two novel transcripts using Sanger sequencing (Supplementary Fig. 2c). Taken together, we have generated a high-quality reference transcriptome for mouse sperm.

To quantify the abundance of spiRNAs, we established standard curves for quantifying the absolute amounts of RNA (using RT-PCR) and for quantifying sperm number (using in vitro synthesized DNA). Three spi-mRNAs, *Rps8*, *Rps6*, and *Rpl11*, which code for 80 S ribosome proteins, were chosen for detection with 154 ± 35 *Rps6* transcripts/sperm, 50 ± 21 *Rps8* transcripts/sperm and 58 ± 21 *Rpl11* transcripts/sperm (Fig. 2e). Since qPCR amplicons are too short to span the entire transcripts, qPCR itself could not distinguish decay intermediates from intact transcripts. However, the sequence reads in sperm were evenly distributed throughout the entire ribosomal protein-encoding transcripts, as seen in RNA-seq in testis (Supplementary Fig. 2d), arguing against the possibility that most of the detected RNAs are decay intermediates. Using the quantification to calibrate the short-read sequencing results, we estimate around 240,000 spiR-NAs per sperm, which supports their potential for biological function.

**spiRNAs include both mRNAs and lncRNAs**. To test the coding potential of spiRNAs during spermatogenesis, we analyzed published ribosome profiling data from mouse testes (Ribo-seq)[60,61]. Ribo-seq detects the RNA fragments protected by their binding to ribosomes from RNase digestion, providing a snapshot of the transcripts that are being translated at a given stage[62,63]. We separated spiRNA species into 2,343 mRNAs and 1,097 lncRNAs using computational approaches to scan homology for sequences and protein domains[64]. We were able to detect RPFs (ribosome protected fragments) enriched at the coding regions of spi-mRNAs, in contrast to the RNA-seq distribution throughout the transcripts (Fig. 3a). Furthermore, the RPFs on spi-mRNAs displayed a three-nucleotide periodicity (Fig. 3b), indicating that elongating ribosomes translate these ORFs. Such signatures of translation were not observed in annotated spi-lncRNAs (Fig. 3b).

We further tested the coding potential of the novel transcripts defined in our studies. For 2,479 novel isoforms from the known loci, 1,538 were annotated as mRNAs. To test whether these novel isoforms are translated in testis, we focused on the isoforms that have altered ORFs. Considering that Ribo-seq, similar to other short-read sequencing methods, also does not maintain transcript integrity, we further narrowed down our analysis to isoforms that include novel exon sequences that contribute to coding regions compared to the annotated RefSeq. We found that RPFs distribute on these novel exon sequences (Fig. 3c, Supplementary Fig. 3), supporting their translation. For the 198 novel transcripts

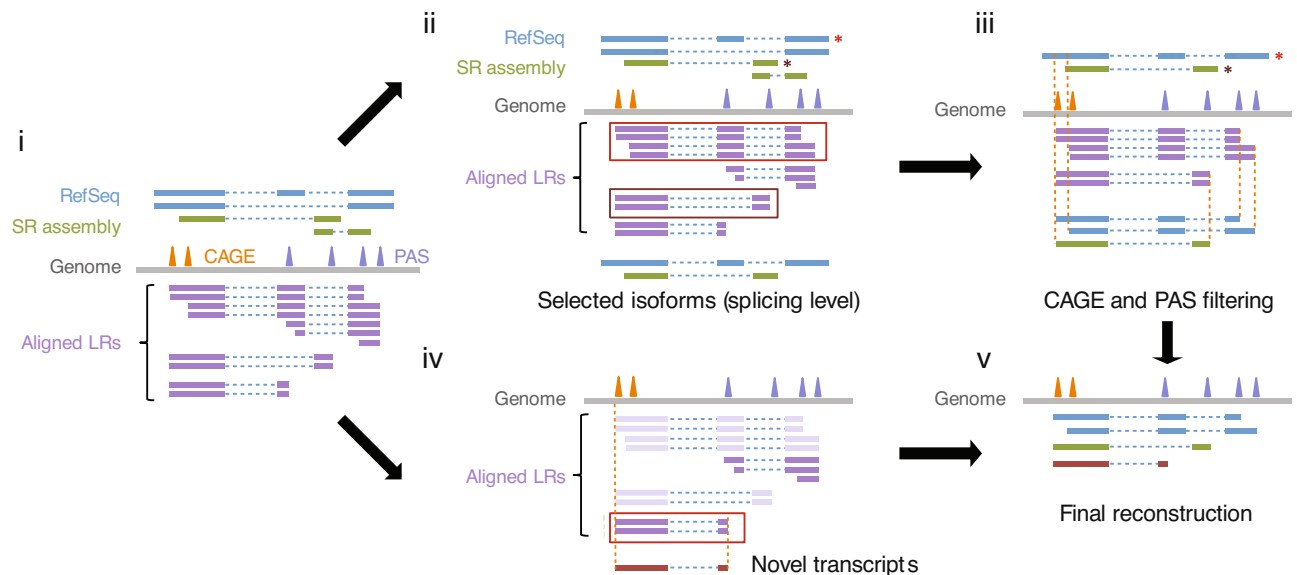

**Fig. 1 Pipeline for identifying intact transcripts in mouse sperm.** The reconstruction of the transcriptome involved five steps. Step **i**: We compared the successfully aligned long reads (LRs) with the RefSeq annotation of mm10 and assembled the short reads (SRs) using StringTie (version 1.3.1c, parameters -c 10 -p 10)[95]. SRs from sperm and testis were assembled separately and the two assemblies were then merged using StringTie again (the "merge" mode). Peaks from CAGE and PAS-seq were defined by AS-peak[96]. Step **ii**: Select isoforms that can be supported by any splicing patterns in RefSeq or SR assembly. It should be noted that we did not impose any restriction at the 5′- or 3′-end at this stage. Step **iii**: Correct the ends of isoforms by CAGE and PAS peaks. Step **iv**: Novel isoforms were defined by selecting LRs that are not supported by either RefSeq or SR assembly. These isoforms must have CAGE and PAS peaks to support their ends. Step **v**: Final assembly contains well annotated intact RNAs in sperm.

from novel loci, we also observed a clear difference between 78 annotated mRNAs and 120 lncRNAs (Fig. 3d, e), similar to what we observed for total transcripts (Fig. 3a, b). In sum, based on their coding potential, we separated spiRNAs into mRNAs and lncRNAs, with the ribosome profiling data supporting that the spi-mRNAs are translated, and are thus likely functional during spermatogenesis.

**spiRNA profiles are distinct from intact transcripts in testis.** Based on gene ontology (GO) enrichment analysis, one of the most significant enrichments of spiRNAs was in mRNAs encoding for 80 S ribosomes (Fig. 4a), which are not required during sperm maturation[19]. This analysis was performed using intact RNA genes from mouse testis as background, thus the enrichment is not due to an existing enrichment of such pathways in the testis. In the effort to provide the background gene list, we performed similar experiments and analysis of the testis long RNAs, using a combination of long-read and short-read sequencing, and detected 11,478 intact transcript species in mouse testis from 5,693 genes which display functional enrichments for spermatogenesis and male gamete generation using all annotated mouse genes as background (Supplementary Fig. 4a). The functional enrichment of translation is contrary to the expectation that spiRNAs are randomly leftover from spermatogenesis[65–67].

Given their functional separation, we tested the length distribution of spiRNAs. Consistent with the observation that sperm total RNAs have a shorter length compared to testis total RNAs (Supplementary Fig. 1a), we found that spiRNAs are significantly shorter than testicular intact RNAs, with their median length (962 nt) being approximately three quarters of the median length of intact testicular RNAs (1,250 nt, Fig. 4b, Wilcoxon rank sum test, $p = 2.0 \times 10^{-200}$). A similar trend was found when we separated spiRNAs into mRNAs and lncRNAs (Supplementary Fig. 4b, $p \leq 5.1 \times 10^{-11}$). Taken together, the

spiRNA profile is distinct from the testicular intact long RNA profile with its shorter length and specific functional enrichment.

**The spiRNA profile is evolutionarily conserved between mice and humans.** To test whether the spiRNA profile is evolutionarily conserved, we performed long-read and short read-sequencing on RNAs from purified human sperm, which have significantly morphologically diverged from mouse sperm (Fig. 4c). Sperm are selected for their species-specific fertilization environments and are considered to be the most diverse cell type known[68]. As illustrated in Fig. 4c, mouse sperm is twice as big as human sperm, with a greater proportion of the length being made up by midpieces. Human sperm has an ovoid head, whereas mouse sperm has an apical hook, allowing mouse sperm aggregates to form a "train" for speed as an adaptation to intense sperm competition[69]. We obtained 14,586,344 subreads, a depth that reached saturation for isoform detection (Supplementary Fig. 1f). Following the same experimental and bioinformatic procedures used for mice, we performed the same 5′ end correction and removal of internal priming using CAGE and PAS data from human testis[70]. In total, we detected 4,100 spiRNAs from 2,205 loci in human with 3,517 mRNAs and 583 lncRNAs.

We found that 30% (562/1,885) of human genes coding for spiRNAs have mouse homologs that also code for spiRNAs, and 46% (641/1,385) of mouse spiRNA genes also have human spiRNA homologs. Of the 17,550 genes shared in both mice and humans, the overlaps between human and mouse spiRNA genes were not random (Fig. 4c, Fisher's exact test, $p = 2.9 \times 10^{-212}$). Among these conserved mRNA genes, using all human spiRNA genes as background to perform the GO enrichment analysis, the mRNAs encoding for protein synthesis were enriched (Fig. 4d), as found in mouse sperm (Fig. 4a). We further analyzed non-ribosomal mRNAs and found that they still share significant overlap (Fisher's exact test, $p = 1.1 \times 10^{-199}$, Supplementary Fig. 4c). These results suggest conservation of the spiRNA

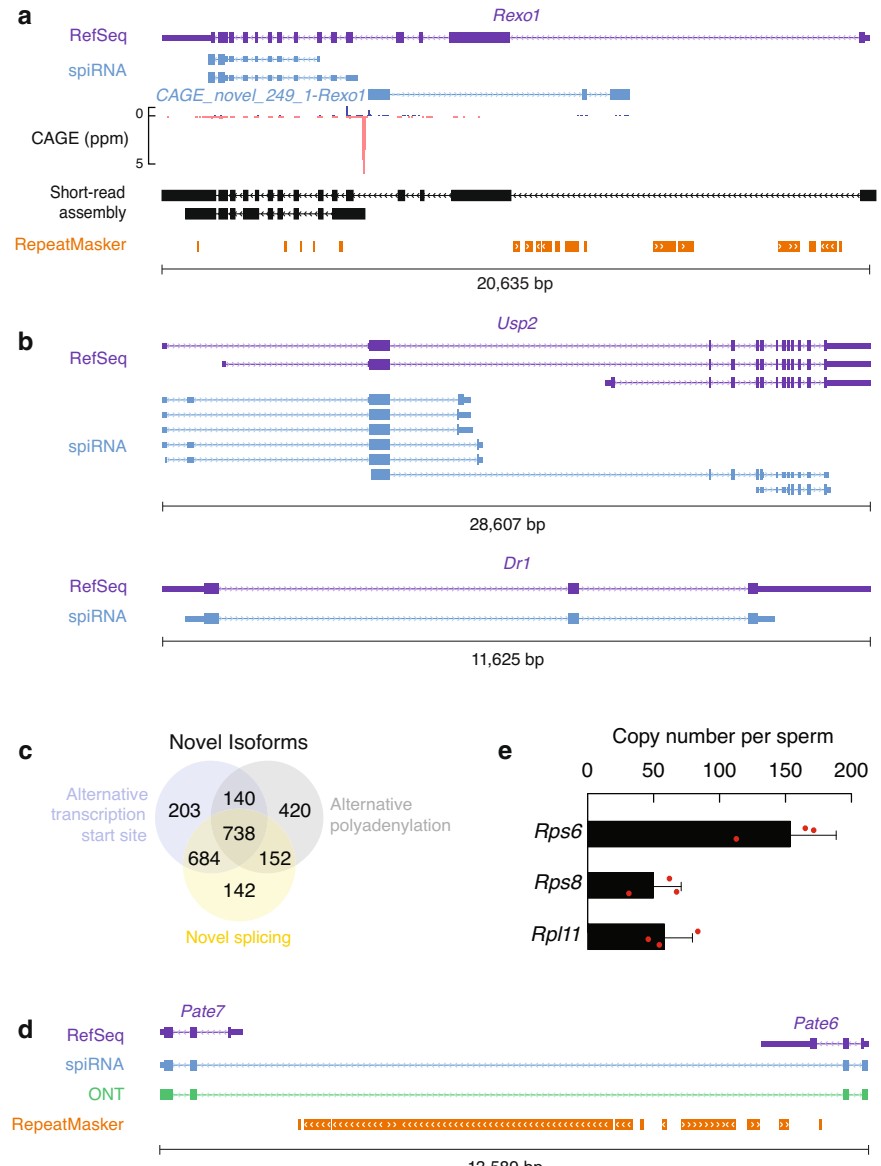

**Fig. 2 Intact long RNA transcripts exist in sperm. a** An example of a novel gene, *CAGE_novel_249_1-Rexo1*, anti-sense to a known gene locus, *Rexo1*, not supported by short-read assembly but detected by PacBio long reads and supported by CAGE, localized in a highly repetitive region of the genome. From top to bottom, RefSeq, spiRNA, CAGE reads (Blue represents Watson strand mapping reads; Red represents Crick strand mapping reads), short-read (SR) assembly, and RepeatMasker. Ppm, parts per million. **b** Two examples of novel transcript structures from known genes, *Usp2* and *Dr1*. From top to bottom, RefSeq and spiRNA. **c** Venn diagrams showing the contribution of alternative transcription start site, alternative splicing, and alternative polyadenylation to the isoforms that differ from RefSeq. **d** A spiRNA spans two neighboring annotated genes, *Pate7* and *Pate6*, which are separated by LINE1 elements. From top to bottom, RefSeq, spiRNA, ONT (Oxford Nanopore Technologies) sequencing reads, and RepeatMasker. **e** Copy number per sperm of spiRNAs, *Rps6*, *Rps8* and *Rpl11*. The RNAs were quantified using in vitro transcribed RNAs as standards. Abundance of transcripts was quantified using RT-qPCR (*n* = 3). Data are mean ± standard deviation. Source data of Fig. 2e are provided as a Source Data file.

repertoire, which likely predates the divergence of rodents and primates 75 million years ago (Fig. 4c)[71].

Similar to the transcript diversity observed in mouse spiRNAs, only 1,327 transcript species were exactly the same as those deposited in RefSeq whereas 10 transcript species were from loci not included in RefSeq, 3 transcripts were anti-sense transcripts to known loci (see Fig. 5a, Supplementary Fig. 5a, for examples) and 2,756 transcript species were novel isoforms from annotated loci with APA contributing the most to the transcript variations (Fig. 5b). Four spiRNAs were found to span two neighboring genes (*FAM153B* and *LOC100507387*, *MICAL2* and *MICALCL*, *CFAP206* and *SLC35A1*, Fig. 5c, Supplementary Fig. 5b). We

found that spiRNAs are significantly shorter (median length: 1,672 nt) than annotated RefSeq (median length: 2,535 nt, Fig. 5d, Wilcoxon rank sum test, $p < 2.2 \times 10^{-16}$), which is consistent with the length distribution of total RNAs purified from human sperm (Supplementary Fig. 1a). Taken together, as in mouse sperm, shorter RNAs are also enriched in human sperm, and we also observed diverse transcripts in human sperm that have not been deposited in RefSeq.

## Discussion

In this study, we demonstrate the existence of intact long RNAs in sperm. The spiRNA profile is evolutionarily conserved between

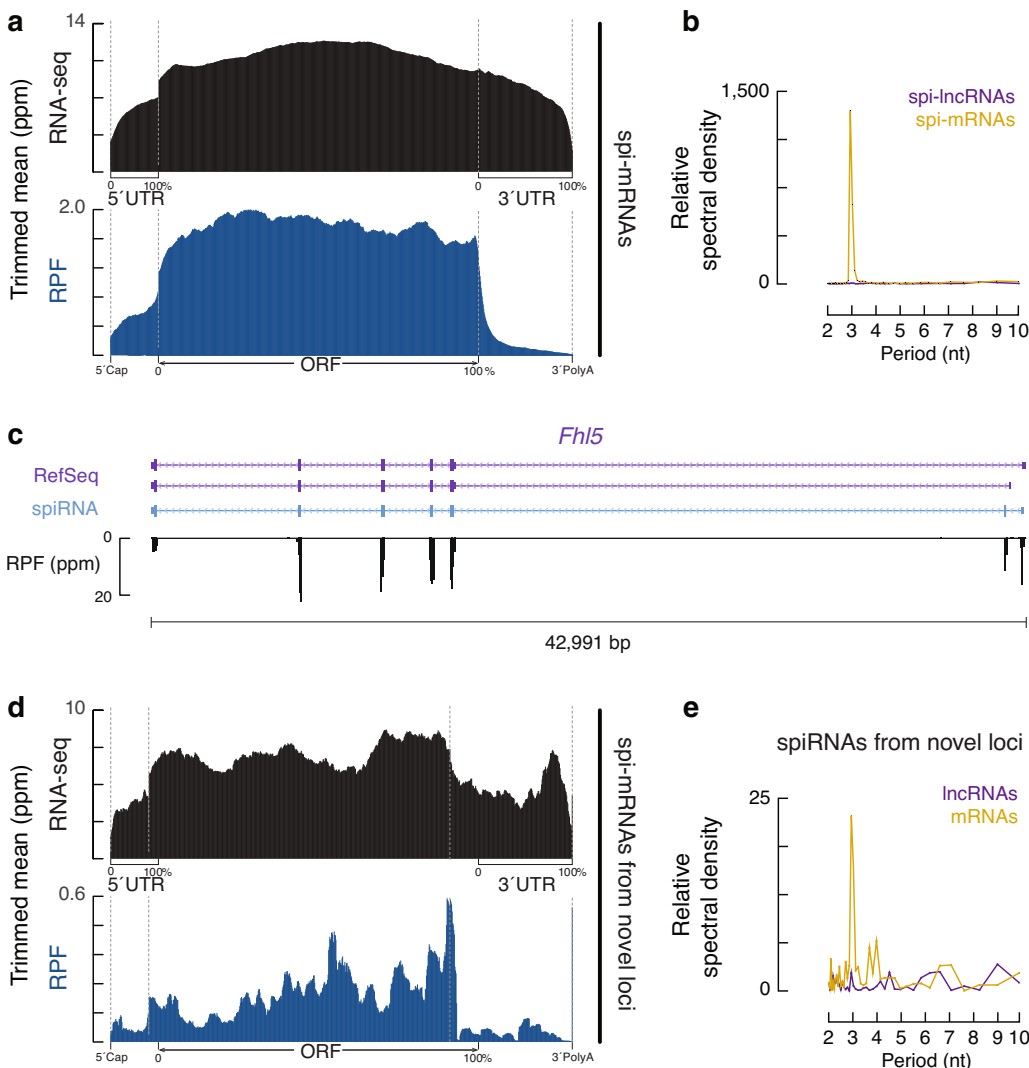

**Fig. 3 spiRNAs include both mRNAs and lncRNAs. a** Aggregated data for RNA-seq signal distribution (*top*) and ribosome occupancy (*bottom*) on 2,343 spi-mRNAs (10% trimmed mean). Signals are aligned to the transcriptional start site (5′cap) and site of polyadenylation (3′PolyA). Signals are further aligned to the ORF regions. RPFs (ribosome protected fragments) across 26–32 nt were used to detect ribosome occupancy. The *x*-axis represents the median length of 5′UTRs, ORFs and 3′UTRs. Ppm, parts per million. **b** Discrete Fourier transformation of the distance spectrum of 5′-ends of RPFs across 2,343 spi-mRNA coding regions (gold) and 1097 spi-lncRNAs (purple) to show nucleotide periodicity. **c** A spiRNA example of novel transcript structures with altered coding regions in comparison to RefSeq annotation. From top to bottom, RefSeq, spiRNA, and RPF (ribosome protected fragment) reads. Ppm, parts per million. **d** Aggregated data for RNA-seq signal distribution (*top*) and ribosome occupancy (*bottom*) on 78 novel spi-mRNAs from intergenic loci (10% trimmed mean). **e** Discrete Fourier transformation of the distance spectrum of 5′-ends of RPFs across coding regions of 78 novel spi-mRNA (gold) and across 120 novel spi-lncRNAs (purple) from intergenic loci to show nucleotide periodicity. Source data of Figs. 3b, 3e are provided as a Source Data file.

mice and humans, and displays an enrichment for protein synthesis functions. This conservation suggests the functional significance of spiRNAs and/or the presence of a conserved mechanism determining the spiRNA repertoire.

We have also developed strategies and standards for defining intact transcripts. We integrated PacBio Iso-Seq long reads with various second-generation sequencing techniques (Fig. 1): (1) Illumina RNA sequencing to correct error-prone long reads and assist in transcript assembly; (2) CAGE sequencing to identify the 5′-ends of intact transcripts; and, (3) PAS sequencing to identify artifactual alternative PolyA sites due to internal oligo-dT priming. We defined 12,794 intact transcripts in mouse sperm and testis, considerably less than the >60,000 total transcripts identified by short-read assembly, demonstrating the parsimony of our transcriptome reconstruction pipeline. Most spiRNAs were not found in RefSeq with the majority of isoforms from known

genes arose from APAs, consistent with observations that proximal PolyA sites are preferred during spermatogenesis[72–74]. The hybrid sequencing and analysis approach used here can be applied to other tissues and organisms to identify intact transcripts and novel isoforms.

The functional enrichment of spiRNAs encoding for the protein components of ribosomes suggests a possible role for these transcripts. There are no intact 80 S ribosomes in sperm, thus spiRNAs are unlikely to be functional until they meet oocyte ribosomes after fertilization. Given that protein synthesis is activated shortly after fertilization[75,76], spiRNAs that encode ribosomal proteins may contribute to quicker and more robust protein synthesis that can direct more resources from the mother to support the pregnancy. The transgenerational role of spiRNAs is suggested by the loss of the effect on offspring after in vitro RNA fragmentation of long RNAs from sperm of fathers treated

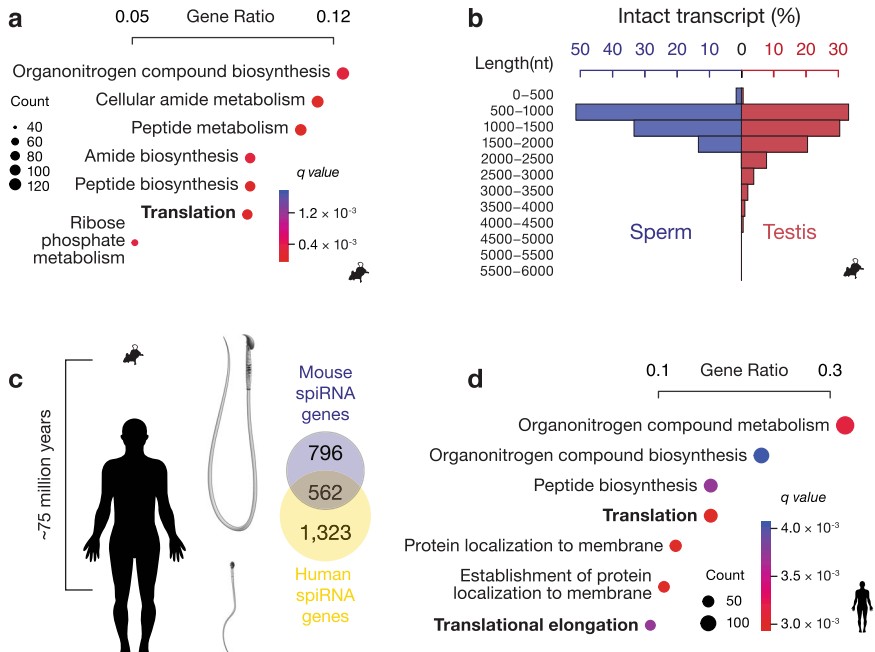

**Fig. 4 The spiRNA profile is evolutionarily conserved. a** GO-term enrichment analysis for biological pathway of mouse spiRNA genes. One-sided hypergeometric test was used to determine the *p*-value. *q* value, adjusted *p*-value using Benjamini-Hochberg correction. The plot displays the top 7 pathways by gene ratio (number of genes related to GO term/total number of genes associated with a GO term in the union of intact RNA genes from mouse sperm and testis). **b** Histogram showing transcript lengths. Blue, spiRNAs. Red, intact testicular long RNAs. **c** Phylogenetic separation of mice and humans as well as the morphology of their sperm and Venn diagrams showing the overlapping gene loci of spiRNAs from mice and humans. **d** GO-term enrichment analysis for biological pathways of conserved spiRNA genes between mice and humans. One-sided hypergeometric test was used to determine the *p*-value. *q* value, adjusted *p*-value using Benjamini-Hochberg correction. The plot displays the top 7 pathways by gene ratio (number of genes related to GO term/total number of human spiRNA genes associated with a GO term). Source data of Fig. 4b are provided as a Source Data file.

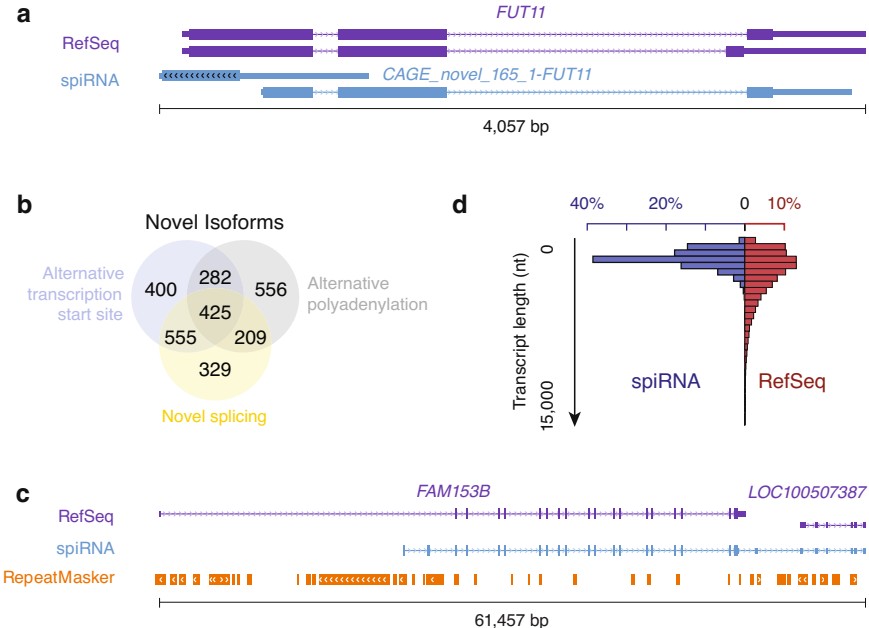

**Fig. 5 Diverse transcripts in human sperm. a** An example of a novel human spiRNA that is antisense to a known gene locus. From top to bottom, RefSeq and spiRNA. **b** Venn diagrams showing the contribution of alternative transcription start site, alternative splicing, and alternative polyadenylation to the isoforms that differ from RefSeq. **c** A spiRNA spans two neighboring genes. From top to bottom, RefSeq, spiRNA and RepeatMasker. **d** Histogram showing transcript lengths in human. Blue, spiRNAs. Red, RefSeq. Source data of Fig. 5d are provided as a Source Data file.

with early mental stress[24]. The early mental stress may predispose their offspring to alter the tendency of the fetus to recruit more resources from the mother. This idea is further supported by data showing that treating sperm with RNase reduces the body weight of the offspring[77]. Considering that the tRNA fragments in sperm not only function transgenerationally[10,11] but are also known to regulate translation[78,79], our finding corroborates the idea that paternally deposited RNAs may alter translation in early embryos[80]. Therefore, our discovery of spiRNAs adds to the concept of the "sperm RNA code" which may be responsive to environmental factors and has the potential to impact the offspring's phenotype.

It remains unclear how environmental signals establish epigenetic changes in the germline. Germ cells are sequestered from somatic cells early in development, and sperm production is protected in a specialized niche. The "Weismann barrier" hypothesis provides a strict demarcation between soma and germ line, restricting the flow of hereditary information from germline to soma and preventing hereditary information flow in the opposite direction[81]. This paradigm is challenged by the identification of transgenerational effects, which indicate routes for soma-to-germline communication. Although we have shown that the sperm transcriptome is distinct from that of the testis, it is possible that the presence of RNAs in sperm is due to those RNAs having higher abundance, lower degradation rates, later transcription activation during spermatogenesis, special subcellular localization, or deposition in epididymis through extracellular vesicles. Therefore, further studies are warranted to elucidate the mechanisms determining the spiRNA profile. An active selection mechanism for spiRNAs, if present, may provide a route for external changes to act on the sperm RNA profile[10,82].

## Methods

**Animals**. C57BL/6 J (Jackson Labs, Bar Harbor, ME, USA; stock number 664) mice were maintained and used according to NIH guidelines for animal care and the University Committee on Animal Resources at the University of Rochester.

**Mouse sperm purification**. After dissecting adult mice, cauda epididymides were excised and placed into 1 ml Whitten's-HEPES buffer pH = 7.3 (100 mM NaCl, 4.4 mM KCl, 1.2 mM $KH_2PO_4$, 1.2 mM $MgSO_4$, 5.5 mM Glucose, 0.8 mM Pyruvic acid, 4.8 mM Lactic acid (hemicalcium), and HEPES 20 mM) at 37 °C. Two additional cuts were made on each epididymis to release sperm. After incubation for 20 min, the sperm suspension was carefully transferred into a 1.5 ml Eppendorf tube, avoiding collection of any contaminating cauda tissue as much as possible. The sperm sample was spun down at 2,655 × g, room temperature, 3 min, washed with $ddH_2O$, and spun down again. To further eliminate somatic cell contamination, the pellet was re-suspended in 500 μl somatic cell lysis buffer (0.1% SDS, 0.05% Triton-X100) at room temperature. After 5 min, the sperm were precipitated by centrifuging at 10,621 × g, room temperature for 3 min. Finally, the liquid was completely removed, and the pellet was ready for sperm RNA purification.

**Human sperm purification**. For conducting these experiments, de-identified normozoospermic (>15 million spermatozoa/ml and >40% motility) semen samples discarded after routine semen analysis were obtained from the Fertility Clinic at the University of Rochester, NY. The study was approved by University of Rochester Medical Center Review Board Protocol 00003599. No informed consent was obtained because IRB considered the samples used in this study to be discarded samples thus IRB exempted. Any existing identification labels were removed and samples were coded. Sperm were prepared using a 90% PureCeption gradient (Sage Biopharma a Cooper Surgical Company, Trumbull, CT). One to two ml of 90% gradient was added to a 15-ml sterile conical tube (Corning), and 1.5–2 ml of semen sample was layered gently on top of the gradient using a sterile transfer pipette. If semen volume was more than 2 ml, we used more than one tube. The sample was centrifuged at 500 × g for 20 min. After centrifugation, seminal plasma and 90% gradient was carefully removed using a transfer pipette leaving a small volume of gradient along with the sperm pellet at the bottom. The sperm pellet was transferred to another clean 15 ml tube, and 5 ml of Quinn's Sperm washing buffer (HEPES-HTF medium with 5 mg/ml human serum albumin) was added, the pellet was resuspended, and centrifuged for 7 min. The sperm pellet was resuspended in 1 ml Sperm wash buffer, and cryopreserved using commercially available Arctic Sperm Cryopreservation Medium (Irvine Scientific, Santa Ana, CA). This medium

contains a mixture of glycerol and sucrose with serum in an isotonic salt solution, and is used commonly in Fertility labs/Sperm bank. Cryopreservation solution (0.33 ml) was added dropwise to the tube containing 1 ml of sperm suspension at room temperature, and mixed gently to obtain (3:1) mixture of sperm and cryo-preservation solution. This mixture was aliquoted in 2–3 cryovials (Corning, NY), secured on an aluminum cryocane, immediately snap frozen in the vapor phase of liquid nitrogen for 30 min, and then plunged into the liquid nitrogen. The coded vials were stored in liquid nitrogen storage tanks specifically designated for research samples in the lab until used.

**Flow cytometry analysis**. We fixed the sperm following the published protocol[83] by adding 16 μl of 50 mM EDTA pH 8 per milliliter of cell suspension (48 μl for 3 ml) so as to obtain a final concentration of 0.8 mM EDTA, and then to fix germ cells slowly adding 3 volumes of ice-cold 100% ethanol using vortex at low speed. The cells are stained with Draq5 DNA stain (DRAQ5™ Fluorescent Probe Solution, Cat# 62251, ThermoFisher Scientific, used at 50 nM final concentration) and data was collected using ImageStream (Luminex Corporation, Austin, TX). Each event was simultaneously imaged in bright field for morphology, in Draq5 fluorescent channel to measure DNA content, and in dark field channel to measure side scatter. By plotting total Draq5 intensity, to differentiate 1n sperm cells and 2n somatic cells, versus Draq5 channel Modulation feature measurement (that measures the intensity range of an image, normalized between 0 and 1) we were able to identify six general population which contained: small debris, cell fragments and larger debris, sperm cells, sperm cells aggregates, somatic cells, and larger aggregates. The analysis was done using IDEAS 6.2 software (Luminex Corporation, Austin, TX).

**Sperm RNA purification**. Before RNA purification, the sperm pellet was lysed in 100 μl lysis buffer (1% SDS, 0.1 M DTT) at 37 °C for 10 min. After incubation, 300 μl Trizol (Thermo Fisher Scientific, Waltham, MA, Catalog #15596018) was added, and the solution was vortexed for 15 seconds, and then placed on an Eppendorf Thermomixer 5436 (Brinkmann Instruments, Inc. Westbury, New York) and vortexed for another 5 min at room temperature. Chloroform (60 μl) was added, and the solution was vortexed for 15 sec, then placed on an Eppendorf Thermomixer 5436 and vortexed for another 3 min at room temperature. The sample was then centrifuged (12,000 g, 4 °C, 15 min), and the liquid phase was transferred to a clean tube. To precipitate RNAs, an equal amount of isopropyl alcohol 2 μl glycogen was added to the solution. After incubation at room temperature for 10 min, the precipitated RNA was pelleted by centrifugation (12,000 g, 4 °C, 10 min). After an additional wash with 75% ethanol, the pellet was dried at room temperature for 5 min. Finally, the RNA was dissolved in 43 μl $ddH_2O$.

To eliminate DNA contamination in the RNA samples, 5 μl of 10X Turbo DNase buffer and 2 μl Turbo DNase (Thermo Fisher Scientific, Waltham, MA, Catalog AM2238) were added to the 43 μl of purified RNA. After incubation at 37 °C for 30 min, 200 μl $ddH_2O$ and 250 μl Acid Phenol: Chloroform (Thermo Fisher Scientific, Waltham, MA, Catalog AM9722) was added, and the sample was mixed thoroughly before centrifugation at 12,000 g at room temperature for 15 min. After centrifugation, the RNA was precipitated by adding 3 volumes of 100% ethanol to 1 volume of supernatant, and 2 μl of glycogen. After a 1 h incubation on ice, the precipitate was pelleted at 15,000 g, 4 °C, 30 min. After an additional wash with 75% ethanol, the pellet was dried at room temperature for 5 min. Finally, the RNA was dissolved in $ddH_2O$ for further library construction.

**Single cell RNA-seq analysis**. We evaluated the sperm purity analytically based on the bulk sperm RNA-seq data and the published single-cell RNA-seq data of epididymis[55]. The epididymis contains various somatic cell types that may pose as probable sources of contamination. The single-cell data was processed using the following steps: (1) normalized the unique molecular identifier (UMI) counts within each cell to make their sum equal to one; (2) grouped the cells into 21 clusters based on the published clustering result[55]; (3) within each cluster, took the median of normalized UMI counts for every gene, which resulted in a consensus expression profile for every cluster. On the other hand, we quantified the expression of all genes in the bulk sperm RNA-seq data and the TPM (transcripts per million) value was calculated using the Salmon algorithm[84] (Supplementary Data 1).

Denote the consensus expression profiles of single-cell RNA-seq data as the matrix X, where rows correspond to genes and columns correspond to 21 clusters. Denote the expression profile of bulk RNA-seq data as Y. We trimmed X and Y to make them share common genes and to ensure none of the genes had zero expression value in Y and all the 21 clusters in X. Then, we performed linear regressions by taking X as the explanatory variables and Y as the observations.

**PacBio Iso-Seq™ library construction and sequencing**. Full-length RNA sequencing libraries (i.e., Iso-Seq™) were constructed according to the recommended protocol by PacBio (PN 101-070-200 Version 05 (November 2017)[85,86]) with a few modifications. Briefly, for analysis of testis, mRNAs were assessed on the Agilent BioAnalyzer or TapeStation, and only preparations with a RIN ≥ 8 were used for sequence analysis. No such RIN requirement was imposed for analysis of sperm RNA. If needed, the ZYMO Research RNA Clean and Concentrator kit (Cat.

# R1015) was used for concentrating dilute samples. RNA preparations considered suitable for IsoSeq typically displayed absorbance ratios of A260/280 = 1.8-2.0 and A260/230 > 2.5.

RNA preparations of similar quality from adult mouse testis and sperm were used for constructing PacBio IsoSeq libraries (SMRT bell libraries). Full-length cDNA was synthesized using the Clontech SMARTer PCR cDNA Synthesis kit (Cat. # 634925) (Clontech, Palo Alto, CA). Approximately 13–15 PCR cycles were required to generate 10–15 μg of ds-cDNA from a 1 μg RNA sample. The library construction steps included: ExoVII treatment, DNA Damage Repair, End Repair, Blunt-end ligation of SMRT bell adaptors, and ExoIII/ExoVII treatment. This procedure resulted in 1.5 microgram of SMRT bell library (i.e., 25–30% yield when starting with 5 microgram of full-length cDNA). Full-length total cDNA was size selected on the ELF SageSciences system (Electrophoretic Lateral Fractionation System) (SageScience), using 0.75% Agarose (Native) Gel Cassettes v2 (Cat# ELD7510), specified for 0.8–18 kb fragments. Fragments were collected in two size ranges, short (0.6–2.5 kb) and long (>2.5 kb). Generation of >2.5 kb fragments required 4 more cycles of amplification and further cleaning. Short and long fragments were combined in an equimolar pool.

For mouse sperm and testis libraries prepared using the PacBio RSII platform, we collected 12 cDNA fractions of various sizes between 0.8 and ~15 kb. These fractions were pooled in such a way as to generate cDNA pools over four size ranges (0.8–2 kb, 2–3 kb, 3–5 kb, and >5 kb) for every sample. Further amplification was needed to generate enough material (for library construction) for the two larger size bins. Additional amplification of the larger size bins resulted in small size byproducts. Therefore, a second size selection (for 3–5 and >5 kb fragments) was performed using an 11 × 14 cm agarose slab gel. Library-polymerase binding was done at 0.01–0.04 nM (depending on library insert size) for sequencing on the PacBio RSII instrument. Diffusion loading was used for the short fragments, and MagBead loading was used for the larger fragments.

Sample cleaning of SageELF fractions and throughout SMRT bell library construction was done following the manufacturer's protocol (PacBio). In brief, fractions were purified with AMPure magnetic beads (0.6:1.0 beads to sample ratio). Final libraries were eluted in 15 μl of 10 mM Tris HCl, pH 8.0. Library fragment size was estimated by the Agilent TapeStation (genomic DNA tapes), and these data were used for calculating molar concentrations. Between 75–125 picomoles of library from each size fraction were loaded onto two SMRT cells for PacBio RS II sequencing. Between 5–8 picomoles of library were loaded (diffusion loading) onto the PacBio Sequel sample plate for sequencing. For optimal read length and output, libraries were sequenced on LR SMRT cell, using 2 h pre-extension, 20 h movies and 2.1 chemistry reagents (for binding and sequencing). All other steps in sequencing were done according to the recommended protocol by the PacBio sequencing calculator and the *RS Remote Online Help* system. For PacBio RS II, one SMRT cell run generated 60–70 thousand reads with a polymerase read length of 13–16 kb. For PacBio Sequel, one SMRT cell run generated 500–70,000 reads with an average polymerase read length of ~27 kb.

## Illumina RNA sequencing.
Strand-specific RNA-seq libraries were constructed following the TruSeq RNA sample preparation protocol as previously described with modifications[56,87]. We treated the total sperm RNAs with DNase first, and then performed the rRNA-depletion with complementary DNA oligos (IDT) and RNase H (Invitrogen, Waltham, MA, USA)[88,89]. Single-end sequencing (126 nt) of libraries was performed on a HiSeq 2000.

## Quantitative reverse transcription PCR (qRT-PCR).
Extracted RNAs were treated with Turbo DNase (Thermo Fisher, Waltham, MA, USA) for 20 min at 37 °C, and were then size-selected to isolate RNA ≥ 200 nt (RNA Clean & Concentrator™−5, Zymo Research) before reverse transcription using All-in-One cDNA Synthesis SuperMix (Bimake, Houston, TX, USA). Quantitative PCR (qPCR) was performed using the ABI Real-Time PCR Detection System with SYBR Green qPCR Master Mix (Bimake). Data were analyzed using DART-PCR[90]. Spike-in RNA was used to normalize RNAs in different samples. Supplementary Table 3 lists the qPCR primers.

## Transcriptome reconstruction and intact mRNA identification.
The PacBio long reads (LRs) were extracted from the bax files using SMRT Analysis pipeline (version 2.3.0). Iso-Seq-classify (default parameters) was then used to identify non-chimeric long reads in which all the 5′ primer sequences, polyA ends, and 3′ primer sequences were present. The long reads which did not satisfy these conditions were filtered out. LoRDEC (version 0.5.3, parameters -k 17 -s 3 -a 50000) was used to perform error correction on the LRs with SRs as input. The corrected LRs were aligned to the mm10 reference genome using GMAP (version 2014-12-24, parameters -t 10 -B 5 -A --nofails -f samse -n1). The aligned LRs with clipping regions longer than 100 nt at either end were filtered out. The SRs were aligned to the mm10 reference genome using HISAT2 (version 2.0.4, parameters --ignore-quals --dta --threads 10 --max-intronlen 150000).

The reconstruction of transcriptome consisted of four main steps (Fig. 1). First, we compared the successfully aligned LRs with the RefSeq (mouse RefSeq annotation of mm10 downloaded from UCSC table browser on May 11th, 2020, human RefSeq annotation of hg38 downloaded March 20th, 2019), and recorded

the isoforms that had full-length LRs. It should be noted that we did not impose any restriction at the 5′ or 3′ end at this stage. In the second step, we assembled the SRs using StringTie (version 1.3.1c, parameters -c 10 -p 10). SRs from sperm and testis were assembled separately, and the two assemblies were then merged using StringTie again (the "merge" mode). The successfully aligned LRs were further compared with the merged SR assembly, and the isoforms which had full-length LRs were recorded. In the third step, we selected the LRs that were supported by CAGE peaks at 5′ ends. We performed clustering on the CAGE supported LRs that were not identified as full-length LRs in the above two steps; this process identified a set of novel isoforms. To reduce false positives, we only retained clusters that contained at least 2 LRs. The novel isoforms were combined with those recorded in the above two steps. In the last step, we inferred the TSS and polyA isoforms at 5′ and 3′ ends. Specifically, we collected the CAGE supported full-length LRs with respect to each isoform. Each collected LR corresponded to a pair of coordinates marking the positions of the CAGE peak and the 3′ end of the LR on the reference genome. By clustering the coordinate pairs, we obtained a detailed annotation of each TSS-polyA isoform.

The identification of intact mRNAs was completed along with the transcriptome reconstruction. Intact mRNAs were the LRs whose 5′ ends were supported by the CAGE peaks or known RefSeq transcripts, and whose 3′ ends were supported by the PAS reads or known RefSeq transcripts.

## General bioinformatics analyses for Illumina sequencing.
Analyses were performed using piPipes v1.4[91]. All data from the RNA sequencing, CAGE, and PAS sequencing were analyzed using the latest mouse genome release mm10 (GCA_000001635.7) and human genome release hg38 (GCA_000001405.27). Generally, one mismatch was allowed for genome mapping. Relevant statistics pertaining to the high-throughput sequencing libraries constructed for this study are reported in Supplementary Data 2.

PAS-seq libraries were analyzed as previously described[56]. We first removed adaptors and performed quality control using Flexbar 2.2 (http://sourceforge.net/projects/theflexibleadap) with the parameters "-at 3 -ao 10 --min-readlength 30 --max-uncalled 70 --phred-pre-trim 10." For reads beginning with GGG including (NGG, NNG or GNG) and ending with three or more adenosines, we removed the first three nucleotides and mapped the remaining sequence with and without the tailing adenosines to the mouse genome using TopHat 2.0.4. We retained only the reads that could be mapped to the genome without the trailing adenosine residues. Genome-mapping reads containing trailing adenosines were regarded as potentially originating from internal priming, and thus were discarded. The 3′ end of the mapped, retained read was reported as the site of cleavage and polyadenylation. We included previously published PAS-seq libraries from wild-type adult mouse testis (GSM1096581)[56] and healthy adult human testis (GSM4030237, GSM4030238, GSM4030239, and GSM4030240)[70] in this analysis.

CAGE was analyzed as previously described[56]. After removing adaptor sequences and checking read quality using Flexbar 2.2 with the parameters of "-at 3 -ao 10 --min-readlength 20 --max-uncalled 70 --phred-pre-trim 10," we retained only reads beginning with NG or GG (the last two nucleotides on the 5′ adaptor). We then removed the first two nucleotides, and mapped the sequences to the mouse genome using TopHat 2.0.4. The 5′ end of the mapped position was reported as the transcription start site. We included previously published CAGE library from wild-type adult mouse testis (GSM1096580)[56] and healthy adult human testis (GSM4030232, GSM4030233, GSM4030234, GSM4030235, and GSM4030236)[70] in this analysis.

Ribo-seq analysis was performed as previously described[61]. Uniquely mapping reads between 26 nt and 32 nt were selected for further analysis. Libraries from different developmental stages were normalized to the sum of reads mapping to mRNA protein-coding regions, assuming that mRNA translation was largely unchanged during spermatogenesis. Nucleotide periodicity was computed as previously described[61,87,92]. We then transformed the distance spectrum using the "periodogram" function from the *GeneCycle* package[93] with the "clone" method. The relative spectral density was calculated by normalizing to the value at the first position. We analyzed published Ribo-seq libraries from mouse wild-type testis at 21 dpp (GSM1234252)[60] and at adult stage (GSE65786)[61].

For RNA-seq reads, the expression per transcript was normalized to the top quartile of expressed transcripts per library as previously described[56]. We analyzed previously published RNA-seq libraries from adult wild-type mouse testis (GSM1088420)[56].

## ONT cDNA Library preparation and sequencing.
The extracted RNA was reverse transcribed using the stand-switching method following the manufacturer's instructions of cDNA-PCR Sequencing Kit (SQK-PCS109, ONT), and full-length transcripts was PCR-amplified using LongAmp Taq 2X Master Mix (M0287, NEB) (95 °C for 15 s, 62 °C for 15 s, 65 °C for 10 min). Eighteen cycles of PCR were performed. The amplified cDNA libraries were prepared by using the cDNA-PCR Sequencing Kit (SQK-PCS109, ONT). Diluting 100 fmol cDNA in 12 μl elution buffer (EB), then, added 1 μl Rapid Adapter (RAP) to the amplified cDNA sample and incubated for 5 min at room temperature. Finally, the prepared cDNA library was loaded into FLO-PRO002 flow cell and sequenced on PromethION Beta.

**ONT analysis**. We aligned the ONT sequencing reads to the mm10 reference genome using minimap2 (version 2.11-r797) with parameter "-x splice -a". Then we compared the uniquely aligned reads with the mouse assembly to identify the full-length long reads, which were the long reads that covered the same sets of exons in an isoform and were consistent at splicing sites. We did not impose any requirement on 5′ or 3′ end match at this step.

**Gene Ontology analysis**. Gene Ontology (GO) analysis was performed using clusterProfiler (version 3.16.0) package from Bioconductor (Release 3.11)[94]. Gene symbols were first converted to Entrez gene id. Enrichment analysis was performed by enrichGO function, using all genes detected in sperm and testis as background. Finally, we used dotplot function to visualize the GO results.

**Nucleotide periodicity**. Nucleotide periodicity was computed as previously described[87,92]. We first aligned the RPFs to each other using 5′-end overlap analysis, and reported the distance spectrum. An annotated ORF was not a prerequisite for this analysis as the distance spectrum of RPFs from mRNAs already showed a 3-nt periodicity pattern. We then transformed the distance spectrum using the "periodogram" function from the *GeneCycle* package[93] with the "clone" method. The relative spectral density was calculated by normalizing to the value at the first position.

**Reporting summary**. Further information on research design is available in the Nature Research Reporting Summary linked to this article.

## Data availability

Sequencing data used in this study have been deposited at the NCBI Gene Expression Omnibus under the accession number GSE137490. We also analyzed published datasets including GSE65786, GSM1234252, GSM1088420, GSM1096581, GSM4030237, GSM4030238, GSM4030239, and GSM4030240, GSM1096580, GSM4030232, GSM4030233, GSM4030234, GSM4030235, and GSM4030236. The data supporting the findings of this study are available from the corresponding authors upon reasonable request. Source data are provided with this paper.

## Code availability

All computational codes used in this study can be obtained from the author upon reasonable request. The codes for PacBio workflow are available at: https://github.com/LiLabZhaohua/PacBioWorkflow

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

## Acknowledgements

We thank D. Amador and the Interdisciplinary Center for Biotechnology Research facility, UR Genomics Research Center, W. Wojciechowski and URMC Flow Cytometry Core for help with the experiments, N. Chen and S. Pitnick for the cartoons of human and mouse sperm, J. L. Weirather for help on data analysis, and members of the Li and Au laboratories for discussions. This work was supported in part by the National Institute of General Medical Sciences [K99/R00HD078482 to X.Z.L.], and by the National Institute of Environmental Health Sciences [pilot project of the center grant P30 ES001247-45 to X.Z.L.]. In addition, this work was supported by the National Human Genome Research Institute [R01HG008759 to K.F.A. and A.W.], an institutional fund of Department of Biomedical Informatics, The Ohio State University [to K.F.A. and A.W.], an institutional fund of Department of Internal Medicine, University of Iowa [to K.F.A. and A.W.], and a startup fund of University of Rochester Center for RNA Biology [to X.Z.L.].

## Author contributions

Y.H.S., A.W. and C.S. analyzed the data with input from K.F.A. and X.Z.L.; Y.H.S., G.S. and R.K.S. performed the experiments with input from K.F.A. and X.Z.L.; K.F.A. and X.Z.L. contributed to the design of the study, and all authors contributed to the preparation of the manuscript.

## Competing interests

The authors declare no competing interests.
