## [Peer Review File · Nature Communications]

Reviewers' comments:

Reviewer #1 (Remarks to the Author):

This manuscript from Sun and colleagues seeks to determine whether long RNA species (e.g., mRNAs, lncRNAs) are carried in the sperm RNA payload as is known to be the case for many small RNA species (e.g., fragments of tRNAs, miRNAs, piRNAs). This is important because the non-genomic sperm payload has been identified as causative in information exchange across generations during generation of the zygote. Also, this serves as one more parameter that could be assessed in the quest to produce sperm in vitro. Moreover, the authors approach this question using long-read RNA-seq, which could provide a first glimpse at the full-length long RNA transcriptome of sperm, which is an exciting new idea. The authors utilize a combination of PacBio and Illumina chemistries to generate new data from cauda epididymal sperm that were prepared to avoid somatic RNA contamination, although the level of validation that the data arose exclusively from sperm were rather limited. Sophisticated combination of their new data with previous 5' and 3' end detection (similar to RACE) powered determination of the termini. Together, the authors detected nearly 11k intact long transcripts, which they name "sperm intact long transcripts" or "SpiLTs." In general, I am not in favor of introducing new terminology or acronyms – in this case, it is unclear why a new term is needed. Regardless, a very interesting result identified in this undertaking is discovery of a number of transcripts that do not exist in the typical mouse RefSeq annotation. Specifically, alternative promoter usage, previously unknown exons, and unique splice variants were detected in these data and powered a novel testis refseq. Similar results were found in human sperm, although the 5'/3' ends are not as precisely mapped due to lack of the CAGE and PAS-Seq. Unfortunately, the authors do not validate that these results actually come from sperm with alternative approaches. This is a key oversight on the part of the authors that draws into question the veracity of their entire study. That is, if the transcripts do indeed arise from sperm, they should be detectable with other methods. Indeed, essentially the rest of the manuscript is devoted towards uncovering where these transcripts arise and how, but that undertaking seems premature given the lack of confirmation that the expression pattern is true. An interesting finding is that the detected intact long mRNAs are biased towards encoding translational machinery, but the significance of this observation is not clear. I am far less impressed than the authors by the finding that the long mRNAs for ribosomal protein subunits supposedly that accumulate in sperm have a greater abundance in sperm from animals subjected to MSUS, a supposed "intergenerational" stress phenomenon that has very poor experimental support. In terms of the origin of long intact sperm transcripts, the low correlations between their levels in 1) testicular and cauda sperm or 2) testicular sperm and round spermatids suggested to the authors that the long RNA payload in sperm are not derived from spermatogenesis. This is an overly speculative conclusion given the lack of wet bench validation of the RNA expression profiles of these intact long RNAs and no empirical experimentation to trace the origin and destination of these transcripts. The use of mouse whole testis transcriptomes from early postnatal life (P0.5 through P26.5) does not appropriately address this issue because the first wave of spermatogenesis is well known to be different from adult spermatogenesis, which occurs in a steady-state. Likewise, the conclusion that because there is a bias towards short mRNAs and lncRNAs in sperm, this must mean there is a post-transcriptional mechanism driving their accumulation is purely speculative based on correlation and lacks any causative support. Further, subsequent bioinformatics analyses demonstrating particular subsets of miRNAs with complementarity to some long transcripts found in sperm are more abundant does not demonstrate co-retention because the authors have not validated that they are present in the same cells at the same time (a problem with bulk/aggregate observations). Overall, while this manuscript studies a very interesting area with a high potential significance for the field, my enthusiasm was significantly tempered by the paucity of validation and empirical evidence combined with significant over-interpretation of the data that ultimately makes this manuscript far too preliminary.

A number of specific criticisms should also be addressed:

1. The occasional focus on MSUS detracts from the quality of this manuscript.
2. Line 29 – references need for spermatid transcriptional quiescence.

3. Line 80 – a number of single-cell studies have been performed with testicular cells and the citation used in support of the proportion of cells is perhaps the least able to establish relative cell numbers because the cells were selected and not profiled without cell sorting. The authors should also consider the long history of literature about cell populations in the testis, which were done way before scRNA-seq was even possible.
4. Line 110 – how was PCR duplication addressed to establish transcript abundance?
5. SpiLTs should be validated with linked-read technology to prove that the transcripts are indeed intact as supposed by the PacBio approach.
6. Line 122 – besides this citation from 1993, are there more contemporary estimates of sperm RNA content? This is a key issue.
7. Lines 129-130 – the detection of ONE transcript expressed by some epididymal cells hardly represents comprehensive validation of lack of somatic cell RNA contamination in sperm RNA preps. Other genes should be tested and other methods must be used to prove that only sperm were present (e.g., FACs/Flow cytometry, microscopy) and only sperm RNA was present (lymphocyte genes, other epididymal genes, etc.). These data must be shown!
8. The degree of biological and technical replication for sperm RNA profiling is not clear.
9. Lines 153-156: this is an extraordinary observation and requires substantial validation to confirm and exclude other explanations, including technical artefacts.
10. Lines 173-175 – it is not clear how this statement is true - claim that long RNA abundance exceeds the amount that could be explained by tethering to the DNA. The authors should provide additional explanation and evidence in support of this conclusion.
11. Line 178 – what is the evidence that SpiLTs “function” in spermatogenesis?
12. Lines 188-190 – the lack of comparable data between human and mouse is a considerable weakness that reduces the impact of inter-species comparison. Essentially, how can a conclusion about the degree of evolutionary conservation be supported if the data types under comparison are different?
13. Line 196 - what is the evidence that these mRNAs actually “encode” the ribosomal components? That is, are the ORFs preserved? This is a reasonable question because of prior indication of novel transcript variants in these data.
14. Line 215 – to claim that there are only two possible explanations comes across as a bit arrogant.
15. Line 219-223 – the human CRISP1 mRNA is detectable in round spermatids in published datasets (reprogenome viewer), which should be considered in deriving an explanation for its apparent sperm accumulation in mice. Indeed, the data presented “suggest” but do not “indicate” (which would require empirical data) epididymal source for this transcript in mouse sperm.
16. Line 236 – validation could help alleviate uncertainty in the statement that SpiLTs “likely localize in the cytoplasmic droplets.” A relevant methodology would be to perform RNA in situ hybridization in a way that would detect these transcripts spatially.
17. Lines 289-291 – this conclusion is supported only by correlation between transcriptomes and not any direct evidence tracing the destiny of RNAs through spermatogenesis. An example method that could shed light would be to use the TU-Tracer system to genetically pulse-label RNAs at various points in spermatogenesis or from somatic cells in the testis or epididymis. Without corroboration, though, this conclusion is simply too speculative.
18. Lines 318-319 – what is the absolute level of each species of RNA? It would seem this explanation would be potentially plausible if the stoichiometry was favorable, but that information is not known.

Reviewer #2 (Remarks to the Author):

The authors of the paper applied the Single-molecule long-read sequencing (pacbio platform) and discovered that there are sperm intact long transcript (SpiLTs) in mature sperm in mouse and human, both showing enriched mRNAs that encoding ribosomal proteins and translational related apparatus, and that SpiLTs are sensitive to early trauma in mouse. These are important findings that will enrich the understanding of “RNA code” in mature sperm and how they respond to paternal

environment/behavior. These results represent the strength of the paper and is novel. However, the second part of conclusion regarding the active post-transcriptional selection of RNAs to retain in sperm spermiogenesis and are co-retained in sperm with miRNAs, is weak, lacking stringent logic, and alternative explanations. I suggest the authors to focus on the first part which I believe is more important, and to make a sturdy work which need more additional analyses. This actually reminds me of a famous saying by the Nobel laureate William G. Kaelin Jr. "Publish houses of brick, not mansions of straw" (Nature. 2017 May 23;545(7655):387.). I put some specific comments as below:

1. first and foremost, the authors should provide a more comprehensive analyses on the length distribution of the SpILTs discovered in both mouse and human, and compare to the length distribution of the rest of mRNAs (that are not intact in the sperm). By doing this, it will reveal whether there is a tendency that the SpILTs are, in general, shorter mRNA that may naturally escape the fragmentation process. I suspect so because ribosome proteins and other translational apparatus are inherently short proteins. And also the authors discovered that there are enrichment of these proteins in both mouse and human and they claim this may represent as conserved selection mechanism, but the alternative explanation is that these ribosomal proteins have many members and they tends to be discovered as a group when performing gene ontology analysis - this may generate a false impression that they are selective enriched, but it could also be a fact that there is simply no selection at all, and they just escape fragmentation effects because they are small, with a similar chance as other short mRNAs – only other mRNA do not group in the pathway/gene ontology analyses. And if this scenario (of negative/random selection) is true, the second part of the paper (so-called active post-transcriptional selection with miRNA) will not be necessary. Again I insist that the author's discovery (especially on the intact mRNA for ribosome proteins) is interesting and important, but it should be carefully interpreted with further analyses. I also recommend he author to read a recent paper relating to analysis bias relating to ribosomal proteins (PLoS Biol 2019 PMID: 31714939).

2. The SpILTs response to mouse MSUS (unpredicted maternal separation combined with unpredictable maternal stress) model is very interesting, and should be analyzed with much more depth. This could be a highlight of the paper and should be comparatively analyzed with the recent relevant paper using the same mouse model (Mol Psychiatry 2018 PMID:30374190). The current analyses are only shown in Fig.2e without any informative description on the specific mRNAs, which is unacceptable.

3. The authors mention in the abstract that the discovery of SpILTs add to the concept of "sperm RNA code". I agree with this and in fact, in a recent paper which coin the term "sperm RNA code" (Nat Rev Endocrinol 2019 PMID: 31235802), it has been discussed extensively how the early embryo "ribosome code" might be regulated by the sperm RNA input, this actually resonate well with the current paper and this could be discussed in the context.

4. The manuscript need more rigor and accuracy, I just picked up some obvious ones as below, the authors should check more thoroughly throughout the manuscript, or this gives a sloppy impression.

4.1 In the abstract, only the number of mouse SpILT is described, but not human, this is not consistent and reads wired. I know the author consider the number of human SpILT is underestimated (line188-190), but the result from the study should be objectively reported in parallel.

4.2 In the abstract Line 23: "rodents and primates" should better be replaced as "mice and humans" as these are the only two species examined in the paper.

4.3 In the same sentence: "The SpILTs profile is evolutionarily conserved between rodents and primates" this should be more specifically described, as the conservation is only discovered for ribosome proteins etc, not for other genes.

4.4 Line 55: in the sentence: "While the transgenerational function of sperm small RNAs has been established7,10,11", In fact, Ref 7,10, 11 mostly deal with "intergenerational" but not "transgenerational" issue.

Reviewer #3 (Remarks to the Author):

Sun et al identified full length transcripts in mouse and human sperms by PacBio long read sequencing with error correction by Illumina short reads. Accurate determinations of 5' and 3' ends are achieved (in mouse) by GAGE- and PAS- sequencing in testis. The experimental and computation methods are well thought out and clearly described, and the resulting sperm and testis transcriptomes appear to be of high quality.

Thousands of sperm intact long transcripts (SpILT) were identified, most of them being novel isoforms seen for the first time. Furthermore, a significant portion of these transcripts are conserved between mouse and human. These results suggest that, despite their transcriptional quiescence, sperms retain a set of transcripts that may have important biological function. The authors further showed that SpILTs are enriched for GO terms related to translation (but not spermatogenesis) and that the abundance of translation related SpILTs is elevated after mental stress. This provided a first hint on the biological functions of these novel transcripts.

In the paragraph starting line 269, the authors regarded the shorter transcripts lengths in the SpLIT population as evidence for active post-transcriptional selection. I do not find this conclusion convincing. Even if these RNAs are randomly retained from precursor cells, it may be natural for longer RNAs may be more likely to be degraded, and such a depletion process can hardly be regarded as an active selection process.

Overall, while the results in this work are not entirely unexpected, the detailed knowledge of the SpILT transcripts will be a very useful resource for further studies on RNA-mediated transgenerational inheritance.

Response to Reviewers' Comments

NCOMMS-19-34062

“Single-molecule long-read sequencing reveals a conserved selection mechanism determining intact long RNA and miRNA profiles in sperm”

We thank the Editor for evaluating our study and the Reviewers for providing constructive comments that have helped us greatly improve our manuscript. In this work, we have used third generation sequencing to answer a long-standing question in the community: do intact long RNAs exist in sperm? We are pleased to see that the Reviewers recognize the importance and novelty of our findings. Based on the various input we have thoroughly revised the manuscript.

We followed the suggestions to remove “*the claims about active post-transcriptional selection of RNAs and their co-retention with miRNAs*” and “*the occasional focus on MSUS which detracts from the study*”. We extended the sections characterizing the sperm intact RNAs (spiRNAs) that harbor repetitive elements, span neighboring genes, or are anti-sense to annotated genes with data represented in the newly added figure panels in both mice and human (Fig. 2 & 5, and Supplementary Fig. 2 & 5). We further addressed comments regarding the sections demonstrating the function of spiRNAs during spermatogenesis by including ribosome profiling data to demonstrate their coding potential in testis and providing further annotation of mRNAs and long non-coding RNAs (newly added Fig. 3, Supplementary Fig. 3). To better reflect the main conclusion, we changed the title to “Single-molecule long-read sequencing reveals a conserved intact long RNA profile in sperm”. We also renamed the sperm intact RNAs from SpILTs to be spiRNAs.

We have incorporated new data to address the Reviewers' concerns: (i) “*the length distribution of the SpiRNAs in mice*” as presented in Fig. 4b and Supplementary Fig. 4b; (ii) We have performed nanopore sequencing as an alternative to validate the spiRNAs defined in our studies, validated spiRNAs with novel isoforms by Sanger sequencing, and tested the somatic contamination by microscopy analysis as shown in Supplementary Fig. 1d, 2b; (iii) we have included the CAGE and PAS data from human testis, and performed the same analysis for human spiRNAs as we did for mouse spiRNAs. The data for human spiRNAs, including their comparison with RefSeq, transcripts harboring repetitive elements that are not assembled by short reads, transcripts spanning neighboring genes, and their length comparison with RefSeq, are included in Fig. 5 and Supplementary Fig. 5. We have also included source data, with data for different figures provided as different sheets within a single Excel file. Below, we provide a detailed point-by-point response to the Reviewers' concerns.

Reviewer #1:

This manuscript from Sun and colleagues seeks to determine whether long RNA species (e.g., mRNAs, lncRNAs) are carried in the sperm RNA payload as is known to be the case for many small RNA species (e.g., fragments of tRNAs, miRNAs, piRNAs). This is important because the non-genomic sperm payload has been identified as causative in information exchange across generations during generation of the zygote. Also, this serves as one more parameter that could be assessed in the quest to produce sperm in vitro. Moreover, the authors approach this question using long-read RNA-seq, which could provide a first glimpse at the full-length long RNA transcriptome of sperm, which is an exciting new idea. The authors utilize a combination of PacBio and Illumina chemistries to generate new data from cauda epididymal sperm that were prepared to avoid somatic RNA contamination, although the level of validation that the data arose exclusively from sperm were rather limited. Sophisticated combination of their new data with previous 5' and 3' end detection (similar to RACE) powered determination of the termini. Together, the authors detected nearly 11k intact long transcripts, which they name “sperm intact long transcripts” or “SpiLTs.” In general, I am not in favor of introducing new terminology or acronyms – in this case, it is unclear why a new term is needed.

Response: We thank the Reviewer for recognizing the significance and novelty of our data. In this revision, we changed the abbreviation from SpILT to spiRNA (sperm intact RNA). We coined the term “spiRNA” to distinguish our work from previous efforts that did not consider the integrity of the transcripts. The assumption in conventional RNA-seq analysis is that all the reads come from the intact transcript. However, this is clearly not the case for sperm RNAs. Therefore, we feel the necessity to stress this key difference in our analysis. We envision similar efforts in other tissues will distinguish the intact pools from degraded pools of transcripts as well.

Regardless, a very interesting result identified in this undertaking is discovery of a number of transcripts that do not exist in the typical mouse RefSeq annotation. Specifically, alternative promoter usage, previously unknown exons, and unique splice variants were detected in these data and powered a novel testis refseq. Similar results were found in human sperm, although the 5'/3' ends are not as precisely mapped due to lack of the CAGE and PAS-Seq. Unfortunately, the authors do not validate that these results actually come from sperm with alternative approaches. This is a key oversight on the part of the authors that draws into question the veracity of their entire study. That is, if the transcripts do indeed arise from sperm, they should be detectable with other methods. Indeed, essentially the rest of the manuscript is devoted towards uncovering where these transcripts arise and how, but that undertaking seems premature given the lack of confirmation that the expression pattern is true.

Response: We thank the Reviewer for the constructive suggestions. Up to date, only a few techniques can capture annotated or novel full-length transcripts (gene isoforms). Here we have validated our results using nanopore sequencing (another form of long-read sequencing), as well as used Sanger sequencing to validate the spiRNAs with novel isoforms (Supplementary Fig. 2c).

An interesting finding is that the detected intact long mRNAs are biased towards encoding translational machinery, but the significance of this observation is not clear. I am far less impressed than the authors by the finding that the long mRNAs for ribosomal protein subunits supposedly that accumulate in sperm have a greater abundance in sperm from animals subjected to MSUS, a supposed “intergenerational” stress phenomenon that has very poor experimental support. In terms of the origin of long intact sperm transcripts, the low correlations between their levels in 1) testicular and cauda sperm or 2) testicular sperm and round spermatids suggested to the authors that the long RNA payload in sperm are not derived from spermatogenesis. This is an overly speculative conclusion given the lack of wet bench validation of the RNA expression profiles of these intact long RNAs and no empirical experimentation to trace the origin and destination of these transcripts. The use of mouse whole testis transcriptomes from early postnatal life (P0.5 through P26.5) does not appropriately address this issue because the first wave of spermatogenesis is well known to be different from adult spermatogenesis, which occurs in a steady-state. Likewise, the conclusion that because there is a bias towards short mRNAs and lncRNAs in sperm, this must mean there is a post-transcriptional mechanism driving their accumulation is purely speculative based on correlation and lacks any causative support. Further, subsequent bioinformatics analyses demonstrating particular subsets of miRNAs with complementarity to some long transcripts found in sperm are more abundant does not demonstrate co-retention because the authors have not validated that they are present in the same cells at the same time (a problem with bulk/aggregate observations). Overall, while this manuscript studies a very interesting area with a high potential significance for the field, my enthusiasm was significantly tempered by the paucity of validation and empirical evidence combined with significant over-interpretation of the data that ultimately makes this manuscript far too preliminary.

Response: We agree with the Reviewer’s concerns that the interpretation of RNA-seq data from the first wave of spermatogenesis should be cautioned as they may not apply to adult stage, and thus have removed the analysis and discussion regarding the origin of the transcripts and co-regulation with the miRNAs. We included the possibility that the shorter transcript length in spiRNA is due to their escaping of random decays in our revised discussion section.

A number of specific criticisms should also be addressed:

1. The occasional focus on MSUS detracts from the quality of this manuscript.

Response: We have toned down the focus on MSUS and removed the part of reanalyzing the previously published MSUS results.

2. Line 29 – references need for spermatid transcriptional quiescence.

Response: We have added the following reference to the main text:

Kierszenbaum, A. L. & Tres, L. L. Structural and transcriptional features of the mouse spermatid genome. *J Cell Biol* **65**, 258-270 (1975).

3. Line 80 – a number of single-cell studies have been performed with testicular cells and the citation used in support of the proportion of cells is perhaps the least able to establish relative cell numbers because the cells were selected and not profiled without cell sorting. The authors should also consider the long history of literature about cell populations in the testis, which were done way before scRNA-seq was even possible.

Response: We thank the Reviewer for pointing this out. We have added a reference characterizing the cell compositions in adult C57/B6 mice (one of the early studies we believe the Reviewer is referring to):

Meistrich, M. L., Bruce, W. R. & Clermont, Y. Cellular composition of fractions of mouse testis cells following velocity sedimentation separation. *Exp Cell Res* **79**, 213-227 (1973).

4. Line 110 – how was PCR duplication addressed to establish transcript abundance?

Response: We have removed the part reanalyzing previously published MSUS results.

5. SpiLTs should be validated with linked-read technology to prove that the transcripts are indeed intact as supposed by the PacBio approach.

Response: Thank you for the suggestion. Although linked-read technology such as 10X Genomics can generate long reads for detecting intact transcripts, it is not widely used because of various technical limits, such as detection of lowly-expressed transcripts. To address the Reviewer's concern, we have validated our transcripts using two orthogonal approaches: (1) For the novel transcripts we detected, we designed primers and submitted the RT-PCR products for Sanger sequencing (Supplementary Fig. 2c); and (2) we used another long-read sequencing method, Oxford Nanopore Technologies (ONT). Although

we do not get a similar read depth as PacBio and the reads have a substantial bias towards shorter fragments, we are still able to validate the intact transcripts detected by PacBio: of the 3,440 spiRNAs in mice, 1,149 had at least one full-length ONT long read.

6. Line 122 – besides this citation from 1993, are there more contemporary estimates of sperm RNA content? This is a key issue.

Response: Sperm RNA content in different mammalian species has been investigated independently by multiple groups. Their results converged to the conclusion that the RNA content per sperm is in the femtogram range. Considering this is a key issue, as noted by the Reviewer, we did a thorough literature review and focused on original articles, rather than review articles, and excluded the articles that did not report the original data for sperm RNA quantification. We have added 6 new references ranging from 1989 to 2018:

- Pessot, C. A. et al. Presence of RNA in the sperm nucleus. *Biochem Biophys Res Commun* **158**, 272-278 (1989).
- Card, C. J., Krieger, K. E., Kaproth, M. & Sartini, B. L. Oligo-dT selected spermatozoal transcript profiles differ among higher and lower fertility dairy sires. *Anim Reprod Sci* **177**, 105-123 (2017).
- Parthipan, S. et al. Spermatozoa input concentrations and RNA isolation methods on RNA yield and quality in bull (*Bos taurus*). *Anal Biochem* **482**, 32-39 (2015).
- Card, C. J. et al. Cryopreserved bovine spermatozoal transcript profile as revealed by high-throughput ribonucleic acid sequencing. *Biol Reprod* **88**, 49 (2013).
- Gòdia, M. et al. A technical assessment of the porcine ejaculated spermatozoa for a sperm-specific RNA-seq analysis. *Syst Biol Reprod Med* **64**, 291-303 (2018).
- Yang, C. C. et al. Identification and sequencing of remnant messenger RNAs found in domestic swine (*Sus scrofa*) fresh ejaculated spermatozoa. *Anim Reprod Sci* **113**, 143-155 (2009).

7. Lines 129-130 – the detection of ONE transcript expressed by some epididymal cells hardly represents comprehensive validation of lack of somatic cell RNA contamination in sperm RNA preps. Other genes should be tested and other methods must be used to prove that only sperm were present (e.g., FACs/Flow cytometry, microscopy) and only sperm RNA was present (lymphocyte genes, other epididymal genes, etc.). These data must be shown!

Response: Considering our method to collect mouse sperm involves having sperm swim up from the cauda epididymis, the major somatic contamination comes from the epididymis, which is the basis for use of an epididymis marker to test for somatic contamination. Although we think one epididymis marker is sufficient to signal for epididymis contamination, we agree with the Reviewer that we should use an independent method to further confirm the lack of contamination of other kinds. Thus, in

the revision, we have included microscopic analyses which demonstrate that after purification, not only were the somatic cells eliminated, but also the cytoplasmic droplets known as Hermes bodies were efficiently removed (Supplementary Fig. 1d). This also further supports our notion that the sperm RNA we detected will travel with the sperm to be deposited into zygotes, rather than be lost as part of cytoplasmic droplets along the journey.

8. The degree of biological and technical replication for sperm RNA profiling is not clear.

Response: We apologize for the lack of clarity. For quantification purposes, there is no doubt that replicates are required to account for biological variations and technical variations. However, we adopted long reads to identify transcripts by full length and not for abundance estimation. Considering the need to account for biological variations, we pooled sperm from over 30 mice and obtained good coverage of the transcript repertoires. To address technical variations, we performed sequencing with multiple flow cells and also employed rarefaction analyses to guarantee the saturation of transcript identification (Supplementary Fig. 1e).”

9. Lines 153-156: this is an extraordinary observation and requires substantial validation to confirm and exclude other explanations, including technical artefacts.

Response: We thank the Reviewer for recognizing our observation as extraordinary. To the best of our knowledge, we are the first to apply this technique to sperm RNAs. Although most of the transcripts we have defined are novel in that they are, either new isoforms from known gene loci, anti-sense transcripts to known loci, or new transcripts from intergenic regions, which are not reported in RefSeq annotation library, it is well-established that RefSeq is merely a reference for the transcriptome and not necessarily the expressed transcript library of a given sample, as demonstrated by past long-read sequencing studies. Based on the following references that have been added in our revision, we were able to annotate the transcripts in greater details, and reported the transcripts spanning two neighboring annotated genes. As we have discussed in the other comments, we have also performed Sanger sequencing and Oxford Nanopore Technologies sequencing to validate our findings:

Rhoads, A. & Au, K. F. PacBio Sequencing and Its Applications. *Genomics Proteomics Bioinformatics* **13**, 278-289 (2015).

Au, K. F. et al. Characterization of the human ESC transcriptome by hybrid sequencing. *Proc Natl Acad Sci U S A* **110**, E4821-30 (2013).

Weirather, J. L. et al. Comprehensive comparison of Pacific Biosciences and Oxford Nanopore Technologies and their applications to transcriptome analysis. *F1000Res* **6**, 100 (2017).

Sharon, D., Tilgner, H., Grubert, F. & Snyder, M. A single-molecule long-read survey of the human transcriptome. *Nat Biotechnol* **31**, 1009-1014 (2013).

Tardaguila, M. et al. SQANTI: extensive characterization of long-read transcript sequences for quality control in full-length transcriptome identification and quantification. *Genome Res* (2018).

10. Lines 173-175 – it is not clear how this statement is true - claim that long RNA abundance exceeds the amount that could be explained by tethering to the DNA. The authors should provide additional explanation and evidence in support of this conclusion.

Response: We infer this conclusion from our results that the transcript abundance of RNA considerably exceeds that of DNA. We appreciate the Reviewer's concern and as we do not believe this conclusion is critical for our study, we have removed this statement.

11. Line 178 – what is the evidence that SpiLTs “function” in spermatogenesis?

Response: The spiRNAs are enriched for genes involved in protein synthesis – a housekeeping role in the cell. We assume that “function” is referring to whether spiRNAs are indeed translated into proteins. As such, we have performed ribosome profiling and demonstrated the translation of the known spi-mRNAs using the spiRNA non-coding RNAs (ncRNAs) as controls (Fig. 3 and Supplementary Fig. 3). Evidence for true function relies on in-depth “wet lab” functional studies which we believe to be beyond the scope of the current manuscript.

12. Lines 188-190 – the lack of comparable data between human and mouse is a considerable weakness that reduces the impact of inter-species comparison. Essentially, how can a conclusion about the degree of evolutionary conservation be supported if the data types under comparison are different?

Response: We agree with the Reviewer that it would be ideal to have comparable data between humans and mice. Thus, in this revision, we have included CAGE and PAS data from human testis and performed the same analysis that was done in mice. We identified 4,100 spiRNAs in humans, further compared them with the known transcripts in RefSeq, and characterized their functional enrichments and length (Fig. 5 and Supplementary Fig. 5). With these new data, our conclusion of a conserved spiRNA profile remains.

13. Line 196 - what is the evidence that these mRNAs actually “encode” the ribosomal

components? That is, are the ORFs preserved? This is a reasonable question because of prior indication of novel transcript variants in these data.

Response: We have included ribosome profiling data which demonstrated a 3nt periodicity on the spi-mRNAs, a signature for active translation. We detected ribosome occupancy on the novel transcripts, which are not reported in RefSeq, with increase the protein coding regions (Fig. 3c and Supplementary Fig. 3). Thus, the ribosome profiling data supports that these ORFs are indeed translated.

14. Line 215 – to claim that there are only two possible explanations comes across as a bit arrogant.

Response: Thank you for pointing out this problem. We acknowledge that we are limited by our current understanding to propose explanations beyond those two, and have removed these sections in this revision.

15. Line 219-223 – the human CRISP1 mRNA is detectable in round spermatids in published datasets (reprogenome viewer), which should be considered in deriving an explanation for its apparent sperm accumulation in mice. Indeed, the data presented “suggest” but do not “indicate” (which would require empirical data) epididymal source for this transcript in mouse sperm.

Response: We thank the Reviewer for providing this reference. We agree that we indeed need to put additional effort into tracing the source of the RNAs, and have thus removed that part from the manuscript.

16. Line 236 – validation could help alleviate uncertainty in the statement that SpiLTs “likely localize in the cytoplasmic droplets.” A relevant methodology would be to perform RNA in situ hybridization in a way that would detect these transcripts spatially.

Response: We have eliminated this part of the results, as discussed in the other responses. In addition, our microscopy analysis shows that the cytoplasmic droplets are efficiently removed (Supplementary Fig. 1d).

17. Lines 289-291 – this conclusion is supported only by correlation between transcriptomes and not any direct evidence tracing the destiny of RNAs through spermatogenesis. An example method that could shed light would be to use the TU-Tracer system to genetically pulse-label RNAs at various points in spermatogenesis or from somatic cells in the testis or epididymis. Without corroboration, though, this conclusion is simply too speculative.

Response: We have eliminated this part of the results, as discussed in the other responses.

18. Lines 318-319 – what is the absolute level of each species of RNA? It would seem this explanation would be potentially plausible if the stoichiometry was favorable, but that information is not known.

Response: We have eliminated this part of the results, as discussed in the other responses.

Reviewer #2:

The authors of the paper applied the Single-molecule long-read sequencing (pacbio platform) and discovered that there are sperm intact long transcript (SpILTs) in mature sperm in mouse and human, both showing enriched mRNAs that encoding ribosomal proteins and translational related apparatus, and that SpILTs are sensitive to early trauma in mouse. These are important findings that will enrich the understanding of “RNA code” in mature sperm and how they respond to paternal environment/behavior. These results represent the strength of the paper and is novel. However, the second part of conclusion regarding the active post-transcriptional selection of RNAs to retain in sperm spermiogenesis and are co-retained in sperm with miRNAs, is weak, lacking stringent logic, and alternative explanations. I suggest the authors to focus on the first part which I believe is more important, and to make a sturdy work which need more additional analyses. This actually reminds me of a famous saying by the Nobel laureate William G. Kaelin Jr. "Publish houses of brick, not mansions of straw" (Nature. 2017 May 23;545(7655):387.). I put some specific comments as below:

Response: We agree with the Reviewer and have eliminated the second part of the conclusion focused on the selection mechanisms. I enjoyed re-reading the article by William G. Kaelin Jr, especially the part: “Both factors encourage reviewers and editors to demand extra experiments that are derivative, tangential to the main conclusion or aimed at increasing impact. And it has always taken more courage to accept a paper than to reject it with suggestions for more experiments. That can create perverse incentives by linking acceptance to a preordained result. I fear that reviewers are especially inclined to ask for more when funding is tight, as it is now.” I appreciate that the Reviewer have asked us to publish less and focus on solidifying our results rather than provide more.

1. first and foremost, the authors should provide a more comprehensive analyses on the length distribution of the SpILTs discovered in both mouse and human, and compare to the length distribution of the rest of mRNAs (that are not intact in the sperm). By doing this, it will reveal whether there is a tendency that the SpILTs are, in general, shorter mRNA that may naturally escape the fragmentation process. I suspect so because

ribosome proteins and other translational apparatus are inherently short proteins. And also the authors discovered that there are enrichment of these proteins in both mouse and human and they claim this may represent as conserved selection mechanism, but the alternative explanation is that these ribosomal proteins have many members and they tends to be discovered as a group when performing gene ontology analysis - this may generate a false impression that they are selective enriched, but it could also be a fact that there is simply no selection at all, and they just escape fragmentation effects because they are small, with a similar chance as other short mRNAs – only other mRNA do not group in the pathway/gene ontology analyses. And if this scenario (of negative/random selection) is true, the second part of the paper (so-called active post-transcriptional selection with miRNA) will not be necessary.

Response: The Reviewer's comments have helped us to greatly improve our manuscript. We have adjusted our analysis on the length distribution accordingly by including a more comprehensive analysis and changing the box plots to histograms (Fig. 4b and Supplementary Fig. 4b) and apply such analysis to human spiRNAs as well (Fig. 5d). Additionally, we have added new discussion on the possibility that the length distribution is merely due to escaping fragmentation. As suggested by the Reviewer, we have removed the second part of the MS on selection mechanisms.

Again I insist that the author's discovery (especially on the intact mRNA for ribosome proteins) is interesting and important, but it should be carefully interpreted with further analyses. I also recommend he author to read a recent paper relating to analysis bias relating to ribosomal proteins (PLoS Biol 2019 PMID: 31714939).

Response: We thank the Reviewer for recognizing the importance of our work and pointing out this useful reference. The only section in our previous version that requires normalization across samples is the quantification of sperm RNAs from MSUS, which has been removed from the revision as it draws attention away from our main conclusion. We will apply the conditional quantile normalization (cqn) method to normalize the estimated raw counts, adapting for the length and CG percentage of the transcripts, as suggested by PLoS Biol 2019 PMID: 31714939 in our future studies.

2. The SpILTs response to mouse MSUS (unpredicted maternal separation combined with unpredictable maternal stress) model is very interesting, and should be analyzed with much more depth. This could be a highlight of the paper and should be comparatively analyzed with the recent relevant paper using the same mouse model (Mol Psychiatry 2018 PMID:30374190). The current analyses are only shown in Fig.2e without any informative description on the specific mRNAs, which is unacceptable.

Response: We apologize for the confusion. We had reanalyzed the data from *Mol Psychiatry* 2018 PMID:30374190 in which they were unable to distinguish spiRNAs from the rest of the decay products and could not find the pathways that are most altered. We have removed this part from the revision, as addressed previously.

3. *The authors mention in the abstract that the discovery of SpILTs add to the concept of “sperm RNA code”. I agree with this and in fact, in a recent paper which coin the term “sperm RNA code” (Nat Rev Endocrinol 2019 PMID: 31235802), it has been discussed extensively how the early embryo “ribosome code” might be regulated by the sperm RNA input, this actually resonate well with the current paper and this could be discussed in the context.*

Response: We appreciate the Reviewer’s suggestion and have included this part in our discussion.

4. *The manuscript need more rigor and accuracy, I just picked up some obvious ones as below, the authors should check more thoroughly throughout the manuscript, or this gives a sloppy impression.*

Response: We appreciate the Reviewer’s time and effort in going through our manuscript carefully and offering the suggestions below.

4.1 *In the abstract, only the number of mouse SpILT is described, but not human, this is not consistent and reads wired. I know the author consider the number of human SpILT is underestimated (line188-190), but the result from the study should be objectively reported in parallel.*

Response: We have included the CAGE and PAS analysis in the human annotation, and have included the numbers in the abstract.

4.2 *In the abstract Line 23: “rodents and primates” should better be replaced as “mice and humans” as these are the only two species examined in the paper.*

Response: We agree with the Reviewer and have fixed the wording as suggested.

4.3 *In the same sentence: “The SpILTs profile is evolutionarily conserved between rodents and primates” this should be more specifically described, as the conservation is only discovered for ribosome proteins etc, not for other genes.*

Response: We have compared the non-ribosomal transcripts and still find a significant overlap in transcripts (Fisher exact test, $p = 9.8 \times 10^{-199}$, Supplementary Fig. 4c). Thus, we have left the wording as is.

4.4 Line 55: in the sentence: “While the transgenerational function of sperm small RNAs has been established^{7,10,11}”, In fact, Ref 7,10, 11 mostly deal with “intergenerational” but not “transgenerational” issue.

Response: Thank you very much for pointing out the difference between “transgenerational” and “intergenerational”. We have adjusted our terminology accordingly, changing from transgenerational to epigenetic.

Reviewer #3:

Sun et al identified full length transcripts in mouse and human sperms by PacBio long read sequencing with error correction by Illumina short reads. Accurate determinations of 5' and 3' ends are achieved (in mouse) by GAGE- and PAS- sequencing in testis. The experimental and computation methods are well thought out and clearly described, and the resulting sperm and testis transcriptomes appear to be of high quality.

Thousands of sperm intact long transcripts (SpILT) were identified, most of them being novel isoforms seen for the first time. Furthermore, a significant portion of these transcripts are conserved between mouse and human. These results suggest that, despite their transcriptional quiescence, sperms retain a set of transcripts that may have important biological function. The authors further showed that SpILTs are enriched for GO terms related to translation (but not spermatogenesis) and that the abundance of translation related SpILTs is elevated after mental stress. This provided a first hint on the biological functions of these novel transcripts.

Response: We appreciate the Reviewer’s comments and have included the novelty recognized by the Reviewers in the discussion.

In the paragraph starting line 269, the authors regarded the shorter transcripts lengths in the SpLIT population as evidence for active post-transcriptional selection. I do not find this conclusion convincing. Even if these RNAs are randomly retained from precursor cells, it may be natural for longer RNAs may be more likely to be degraded, and such a depletion process can hardly be regarded as an active selection process.

Response: We agree with the Reviewer’s concern. Based on the Editor’s and the Reviewers’ suggestion, we have removed the part on active selection in this revision.

Overall, while the results in this work are not entirely unexpected, the detailed knowledge of the SpILT transcripts will be a very useful resource for further studies on RNA-mediated transgenerational inheritance.

In summary, our study presents a new finding regarding the sperm transcriptome. In the revision, we have included 6 new figure panels in the existing figures and added the Figures 3 & 5 and Supplementary Figures 3 & 5. We envision this manuscript will stimulate new avenues of research aimed at uncovering their function and biogenesis of spiRNAs. We appreciate your favorable consideration of publishing our work in *Nature Communications*.

REVIEWER COMMENTS

Reviewer #1 (Remarks to the Author):

This is a revised manuscript from Sun and colleagues that argues the reproducible presence of full-length mRNA transcripts [called sperm intact RNAs (spiRNAs) by the authors] as normal payload in mammalian sperm. The authors have responded appropriately to many of my concerns with the initial manuscript and this revision is considerably more focused, explores those focused data more deeply, and is generally improved over the initial version. I continue to maintain that the detection of full-length mRNA payloads in sperm is an intriguing and potentially very important finding, but is only valid IF the results indeed arise from sperm. Hence, one of my primary concerns with the initial manuscript was that the authors had not demonstrated purity of their input sperm samples. Essentially, since mRNA abundance in sperm is, by the authors own admission (lines 113-115), low, this means that any very low level of contamination with epididymal somatic cells can vastly influence the results. The initial manuscript included one purity metric, mRNA levels of the smooth muscle gene the smooth muscle gene *Myh11*. In the revision, the authors continue to rely upon detection of this gene and indicate that *Myh11* is "a marker for epididymis contamination" (lines 120-121), citing a 2010 Cell paper from Ollie Rando's group in support. It should be noted, though, that there is now very definitive data demonstrating that *Myh11* is only robustly expressed by the small subset of epididymal smooth muscle cells (see Figures 1-2 and associated text from a BioRxIV manuscript from Ollie Rando's group that was posted January 2020 - <https://doi.org/10.1101/2020.01.24.918979>). In contrast, numerous other mRNAs label other, more abundant epididymal cell types, including principle cells, clear cells, basal cells, immune cells and some stromal cells. To actually address this criticism with new data, though, the authors presented three panels of phase contrast micrographs of post-purification sperm samples (Fig. S1d) that, (in total) show all/part of 15 sperm. Unfortunately, while those data are helpful, they are far from convincing that contamination is absent above 1/16 cells and do not support that authors contention that somatic cells have been eliminated and the resulting sperm samples are "ultra-pure" (line 126). Again, demonstration that *Myh11* levels are below the LOD (lines 122-123) provides some reassurance that smooth muscle contamination is low (if not absent), but does not exclude contamination by other epididymal somatic cell types. Multiple quantitative measures (e.g., detecting multiple markers from each epididymal somatic cell type, counting cells on micrographs or performing flow cytometry) are necessary to prove this point convincingly. As noted in my critique of the initial manuscript, without careful and thorough validation of the input sample, my confidence in the conclusions is not high.

Reviewer #2 (Remarks to the Author):

The authors have satisfactorily addressed my comments.

Reviewer #3 (Remarks to the Author):

The authors have removed the section found to be unconvincing in my previous review. I am now happy with the paper.

Response to Reviewers' Comments

NCOMMS-19-34062A

“Single-molecule long-read sequencing reveals a conserved selection mechanism determining intact long RNA and miRNA profiles in sperm”

We are very pleased that that two of the Reviewers are now satisfied with our revisions and that Reviewer #1 has only one remaining concern. Below, we provide a detailed point-by-point response to the Reviewer#1' concern.

Reviewer #1 (Remarks to the Author):

This is a revised manuscript from Sun and colleagues that argues the reproducible presence of full-length mRNA transcripts [called sperm intact RNAs (spiRNAs) by the authors] as normal payload in mammalian sperm. The authors have responded appropriately to many of my concerns with the initial manuscript and this revision is considerably more focused, explores those focused data more deeply, and is generally improved over the initial version. I continue to maintain that the detection of full-length mRNA payloads in sperm is an intriguing and potentially very important finding, but is only valid IF the results indeed arise from sperm. Hence, one of my primary concerns with the initial manuscript was that the authors had not demonstrated purity of their input sperm samples. Essentially, since mRNA abundance in sperm is, by the authors own admission (lines 113-115), low, this means that any very low level of contamination with epididymal somatic cells can vastly influence the results.

The initial manuscript included one purity metric, mRNA levels of the smooth muscle gene the smooth muscle gene Myh11. In the revision, the authors continue to rely upon detection of this gene and indicate that Myh11 is “a marker for epididymis contamination” (lines 120-121), citing a 2010 Cell paper from Ollie Rando's group in support. It should be noted, though, that there is now very definitive data demonstrating that Myh11 is only robustly expressed by the small subset of epididymal smooth muscle cells (see Figures 1-2 and associated text from a BioRxIV manuscript from Ollie Rando's group that was posted January 2020 - <https://doi.org/10.1101/2020.01.24.918979>). In contrast, numerous other mRNAs label other, more abundant epididymal cell types, including principle cells, clear cells, basal cells, immune cells and some stromal cells. To actually address this criticism with new data, though, the authors presented three panels of phase contrast micrographs of post-purification sperm samples (Fig. S1d) that, (in total) show all/part of 15 sperm. Unfortunately, while those data are helpful, they are far from convincing that contamination is absent above 1/16 cells and do not support that authors contention that somatic cells have been eliminated and the resulting sperm samples are “ultra-pure”

(line 126). Again, demonstration that *Myh11* levels are below the LOD (lines 122-123) provides some reassurance that smooth muscle contamination is low (if not absent), but does not exclude contamination by other epididymal somatic cell types. Multiple quantitative measures (e.g., detecting multiple markers from each epididymal somatic cell type, counting cells on micrographs or performing flow cytometry) are necessary to prove this point convincingly. As noted in my critique of the initial manuscript, without careful and thorough validation of the input sample, my confidence in the conclusions is not high.

Response: We are glad to see Reviewer #1 shares our view that the purity of sperm is critical for sperm RNA studies. However, I apologize that we were not clear enough in the previous version of the MS and that Reviewer #1 may have misunderstood the source of the contamination. During sperm collection, we cut the cauda epididymis and let the sperm swim out to collect them. The potential contamination would only come from cutting the epididymis when releasing the sperm. The resulting lesion would release somatic RNAs from a mixture of different types of cells from the epididymis. Considering that *Myh11* is both broadly and highly expressed in the muscular sheath of the cauda epididymis, according to Table S2 of <https://doi.org/10.1101/2020.01.24.918979> referred to by Reviewer #1, where sperm are released and collected, we should be able to detect *Myh11* expression if contamination arises from cutting the epididymis. Thus, we, and the previous study from the Rando lab, chose *Myh11* as a marker for mouse sperm contamination.

We would also like to apologize for the misunderstanding that we only looked at 15 sperm for our microscopy validations. We only showed representative figures of the sperm when in fact, we actually looked at over 1,000 sperm for these studies. With regard to the flow cytometry results, we have analyzed sperm in this way before and, in our experience, the micrographs are reliable for identifying the cells, especially pure sperm which do not display a unique morphology during flow because they break apart.

We have revised our results section on Page6-7 with changes highlighted to include new references and improve clarity on the source of the contamination and the microscopy analysis.

Reviewer #2 (Remarks to the Author):

The authors have satisfactorily addressed my comments.

Reviewer #3 (Remarks to the Author):

The authors have removed the section found to be unconvincing in my previous review. I am now happy with the paper.

In summary, we believe that we have conducted stringent testing of sperm purity to fully validate our characterization of the full-length intact mRNA transcripts in sperm. We envision this manuscript will stimulate new avenues of research aimed at uncovering their function and biogenesis of spiRNAs. We appreciate your favorable consideration of publishing our work in *Nature Communications*.

REVIEWER COMMENTS

Reviewer #4 (Remarks to the Author):

I have read the manuscript and I am in full agreement with Reviewer 1. The purity of the samples is absolutely critical to support the core claims of the manuscript. Without a convincing validation with an array of controls, I would not be supportive of its publication.

REVIEWERS' COMMENTS

Reviewer #1 (Remarks to the Author):

This is a second revision to a manuscript from Sun and colleagues that adds two new experiments that provide more persuasive evidence that the input sample is mostly sperm. Further, the authors appropriately characterize their sample and containing little to no somatic contamination in the text. I have no further concerns with this paper.